# Time-restricted feeding normalizes hyperinsulinemia to inhibit breast cancer in obese postmenopausal mouse models

Manasi Das[1,2], Lesley G. Ellies[3,4], Deepak Kumar[1,2], Consuelo Sauceda[1,2], Alexis Oberg [1], Emilie Gross[1,2], Tyler Mandt[5], Isabel G. Newton[5], Mehak Kaur[2], Dorothy D. Sears[2,4,6,7] & Nicholas J. G. Webster [1,2,4✉]

Accumulating evidence indicates that obesity with its associated metabolic dysregulation, including hyperinsulinemia and aberrant circadian rhythms, increases the risk for a variety of cancers including postmenopausal breast cancer. Caloric restriction can ameliorate the harmful metabolic effects of obesity and inhibit cancer progression but is difficult to implement and maintain outside of the clinic. In this study, we aim to test a time-restricted feeding (TRF) approach on mouse models of obesity-driven postmenopausal breast cancer. We show that TRF abrogates the obesity-enhanced mammary tumor growth in two orthotopic models in the absence of calorie restriction or weight loss. TRF also reduces breast cancer metastasis to the lung. Furthermore, TRF delays tumor initiation in a transgenic model of mammary tumorigenesis prior to the onset of obesity. Notably, TRF increases whole-body insulin sensitivity, reduces hyperinsulinemia, restores diurnal gene expression rhythms in the tumor, and attenuates tumor growth and insulin signaling. Importantly, inhibition of insulin secretion with diazoxide mimics TRF whereas artificial elevation of insulin through insulin pumps implantation reverses the effect of TRF, suggesting that TRF acts through modulating hyperinsulinemia. Our data suggest that TRF is likely to be effective in breast cancer prevention and therapy.

[1] VA San Diego Healthcare System, San Diego, CA, USA. [2] Department of Medicine, Division of Endocrinology and Metabolism, University of California San Diego, La Jolla, CA, USA. [3] Department of Pathology, University of California San Diego, La Jolla, CA, USA. [4] Moores Cancer Center, University of California, San Diego, La Jolla, CA, USA. [5] Department of Radiology, University of California, San Diego, La Jolla, CA, USA. [6] Department of Family Medicine and Public Health, Division of Preventive Medicine, University of California San Diego, La Jolla, CA, USA. [7] College of Health Solutions, Arizona State University, Phoenix, AZ, USA. ✉email: nwebster@health.ucsd.edu

Obesity is a serious health threat worldwide, with the prevalence of obesity and metabolic syndrome increasing to epidemic levels over the last three decades, particularly in Western countries[1,2]. Obesity and its associated metabolic deregulation is a strong risk factor for type 2 diabetes, heart, and kidney disease,[3–6] and, importantly, at least 13 types of cancer including postmenopausal breast cancer, colorectal cancer, endometrial cancer, esophageal adenocarcinoma, and gallbladder cancer[7–10]. Recent studies have revealed that obesity and breast cancer are strongly associated[11]. Because breast cancer is the most common cancer and the second leading cause of cancer death among women in developed countries[12], understanding how obesity impacts this disease has important public health implications. The impact of obesity on breast cancer is highly complex depending on menopause status, receptor expression, inflammation, and other factors[11]. For example, the effect of obesity on breast cancer risk depends on menopause status. In premenopausal women, studies report that obesity either reduces or increases breast cancer risk, but obesity is consistently reported to be associated with higher breast cancer risk and poorer breast cancer outcomes in postmenopausal women[11]. However, like other metabolic dysregulation, the mechanisms underlying the impact of obesity on breast cancer incidence, morbidity, mortality are not fully understood and studies to establish a mechanistic link are ongoing. A number of studies implicate hyperinsulinemia and insulin resistance as a potential driver in obesity-driven breast cancer[13,14]. Indeed, a Mendelian randomization analysis showed a positive association between genetically predicted fasting insulin with breast cancer risk and suggest that genetically determined obesity and insulin-related traits play an important role in the etiology of breast cancer[15]. Furthermore, insulin resistance was associated with breast cancer in obese, but not normal weight, premenopausal women, or postmenopausal women[14,16]. Prospective data from the Women's Health Initiative study, however, indicate that overweight women who are metabolically healthy and have normal insulin sensitivity do not have an increased risk for breast cancer, whereas normal weight or obese women with insulin resistance are at risk, suggesting that insulin resistance, rather than obesity per se, is the driving factor[17]. Therefore, focusing on improving the insulin resistance of obese postmenopausal women may reduce their risk of breast cancer.

Obesity disrupts the circadian clocks in the brain and peripheral tissues that generate 24 h rhythms in gene expression and diurnal behaviors[18,19]. Menopause in women also causes circadian sleep disturbances[20], likely owing to the decrease in ovarian steroids, as steroids have been shown to alter sleep rhythms in rats[21,22]. Such circadian clock disruption can lead to the development of insulin resistance and metabolic syndrome, and can predispose individuals to a number of chronic diseases including cancer[23,24]. Circadian clock disruption has been reported in tumor cells and this is thought to promote tumor growth owing to the dysregulation of key cell cycle and tumor suppressor genes that are under clock control[25]. Consequently, treatments that improve circadian rhythms in tumors slow cell cycle progression and reduce proliferation and tumor growth[26]. The mechanistic links between obesity, insulin resistance, menopause, and circadian rhythm disruption in cancer progression, however, are poorly understood. Given the ability of the circadian clock to directly control several pathways that are crucial for both glucose metabolism and tumorigenesis, reinforcing robust rhythms may provide an effective strategy for cancer prevention and requires further investigation.

Not surprisingly, weight loss in obese individuals by caloric restriction or fasting leads to beneficial effects on metabolism and on reducing cancer growth in both mice and humans[27–30].

Despite promising outcomes, weight-loss regimens have not been widely adopted in cancer prevention, survivorship, or during therapy as they cause hunger and irritability, and can have challenging exercise, calorie-counting, and/or menu-planning designs, all of which limit long-term adherence[30–32]. An alternative approach for enhancing metabolic health is to restrict the timing of calorie intake rather than the quantity[33]. Epidemiological and preclinical studies have shown that prolonged nightly fasting (the period between the last meal at night and the first meal of the day) is associated with reduced breast cancer risk and recurrence[28,33–36]. Experimentally, time-restricted feeding (TRF) is a form of intermittent fasting in which food intake is limited to a particular number of hours per day, typically 6–12 h and in alignment with circadian rhythms[28,33,34,37]. Studies in mice have demonstrated that TRF of a high-fat diet (HFD) protects against and can reverse adverse obesity-related dysfunction, including metabolic changes, inflammation, and weight gain, and restores normal circadian rhythms when the food intake occurs during the normal active phase[19,38–40]. Similar beneficial effects have been observed in small pilot studies in humans. One study involving 23 obese individuals, predominantly women, reported that time-restricted eating (TRE, the human equivalent of TRF in animals) reduced bodyweight and systolic blood pressure[41]. TRE improved cardiometabolic health in 19 subjects, men and women, with metabolic syndrome receiving standard medical care[42]. Another study in 20 obese subjects, again predominantly women, reported that TRE causes weight loss and alters body composition with significant reductions in visceral fat mass[43]. TRE also improved insulin sensitivity and blood pressure in two small studies of men with prediabetes ($N = 8$)[44,45]. So, beneficial metabolic effects are seen in both sexes, which is consistent with studies in obese mice[38,40]. Although promising benefits have been observed in such pilot studies, appropriately powered, randomized controlled trials are lacking[41–43,46]. Importantly for compliance, mouse TRF studies show that adherence to the TRF schedule Monday to Friday is sufficient for full metabolic benefit, indicating that translation of this intervention modality to human subjects may allow for some behavioral flexibility on the weekend[38]. Thus, TRE is an attractive intervention to reduce insulin resistance and restore normal circadian rhythms, as it requires neither a change in diet nor physical activity. Fasting occurs primarily during sleep, thus reducing the hunger and irritability associated with daytime fasting and thereby making long-term compliance more feasible[30,41,46].

Given the strength of data in the literature, we investigated the beneficial effects of TRF to reduce or prevent breast cancer using mouse models of postmenopausal obesity. In this work, we demonstrate that TRF abrogates the obesity-enhanced mammary tumor growth in two orthotopic models in the absence of calorie restriction or weight loss. TRF also reduces breast cancer metastasis to the lung. Furthermore, TRF delays tumor initiation in a transgenic model of mammary tumorigenesis prior to the onset of obesity. Notably, TRF increases whole-body insulin sensitivity, reduced hyperinsulinemia, restored diurnal clock gene expression rhythms in the tumor and non-tumor tissues, and attenuates insulin signaling. Finally, we provide evidence that inhibition of insulin secretion mimics the TRF effect, whereas artificial elevation of insulin reverses it, suggesting that TRF acts by modulating hyperinsulinemia.

## Results

**TRF improves obesity-driven metabolic dysfunction without altering caloric intake in postmenopausal mouse models.** We first evaluated the effects of TRF on metabolic dysfunction and breast cancer in two models of postmenopausal obesity induced

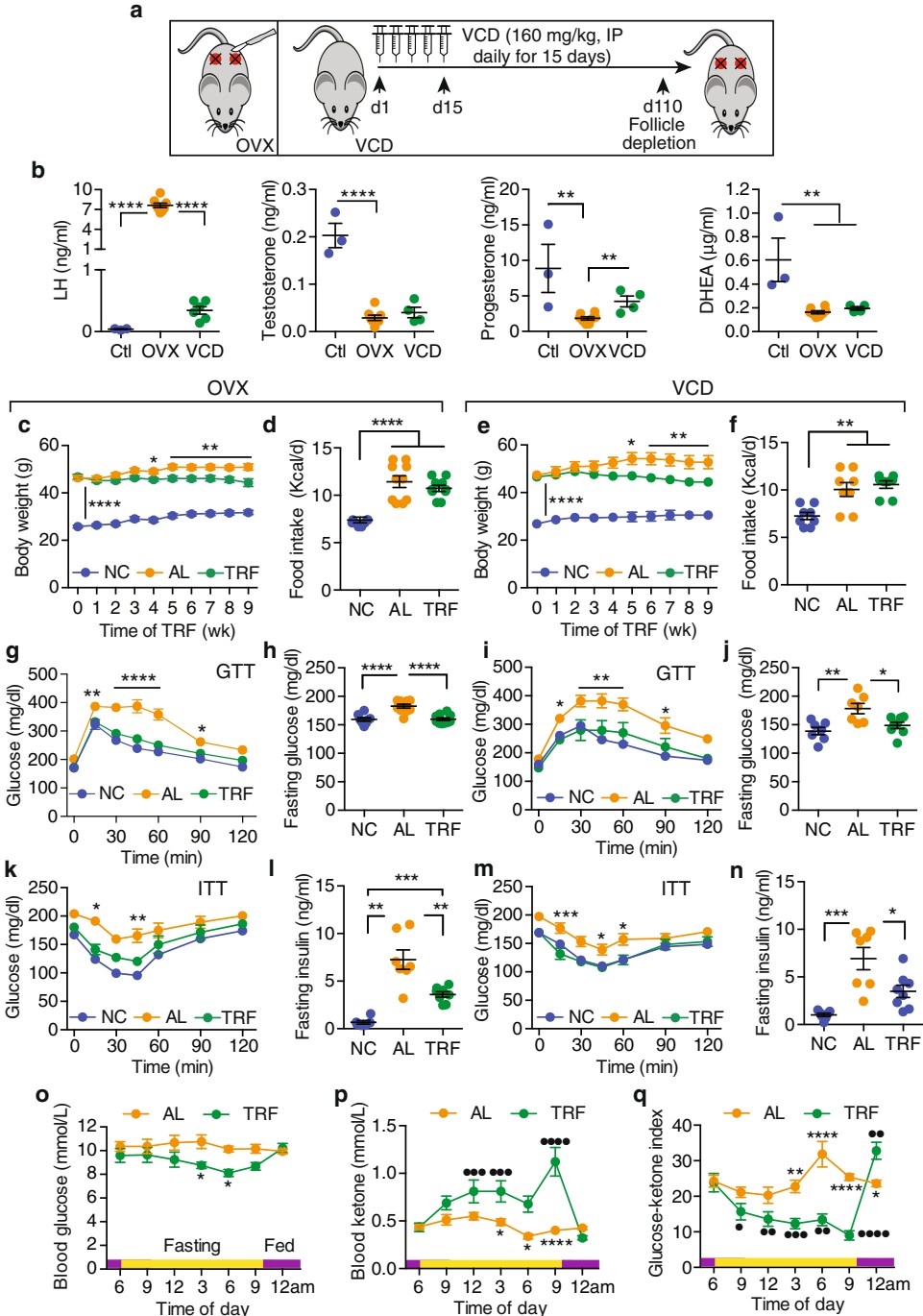

either by bilateral ovariectomy (OVX) or by follicle depletion using 4-vinylcyclohexene diepoxide (VCD) administration (Fig. 1a). Ovariectomy makes rodents susceptible to HFD-induced disruption of circadian rhythms, an effect that can be corrected with exogenous estrogen[47,48]. As expected, plasma levels of luteinizing hormone (LH) were greatly elevated in OVX and moderately elevated in VCD mice (Fig. 1b). Levels of progesterone, testosterone, and dehydroepiandrosterone (DHEA) were lower in both OVX and VCD models, consistent with a postmenopausal state (Fig. 1b). The mice were fed ad libitum HFD for 10 weeks before random assignment to continued ad libitum HFD (AL) or time-restricted HFD (TRF) groups (Supplementary Fig. 1a). A control group of mice was fed ad libitum normal chow (NC). These control mice remained lean

throughout the study in both the OVX and VCD models (Fig. 1c, e, Supplementary Fig. 1b, d). Both AL groups gained weight throughout the study period, whereas both TRF groups showed stabilization of bodyweight (Fig. 1c, e), which was reflected in a difference in bodyweight at euthanasia (Supplementary Fig. 1b, d). Food intake did not differ between groups (Fig. 1d, f), confirming that there was no caloric restriction in the TRF model.

Similar to what we have reported in female OVX mice[40] and others have reported in male mice[19,38], 7 weeks of TRF resulted in improved glucose tolerance compared with the AL group in both the OVX and VCD models (Fig. 1g, i, Supplementary Fig. 1c, e). Terminal fasting blood glucose (FBG) levels were elevated in the AL groups, but terminal FBG levels in the TRF groups were indistinguishable from those in NC mice (Fig. 1h, j). In addition,

**Fig. 1 TRF reduces obesity and improves metabolic dysregulation in obese HFD-fed postmenopausal mouse models. a** Schematic of female postmenopausal mouse models showing bilateral ovariectomy (OVX) or daily 4-vinylcyclohexene diepoxide (VCD) treatment over 15 days in C57BL/6 mice. **b** Levels of luteinizing hormone (LH), testosterone, progesterone, and dihydroandrosterone (DHEA) in plasma from OVX and VCD mice compared with control (Ctl) ovary intact mice (number of mice $n = 3$ for Ctl, $n = 8$ for OVX, $n = 4$–6 for VCD; **$p < 0.0$, ****$p < 0.0001$ by one-way ANOVA with Tukey's multiple comparisons test). Ctl mice are colored blue, OVX mice orange, and VCD mice green. **c, e** Weekly body weights of OVX and VCD mice on ad libitum normal chow (NC), ad libitum high-fat diet (AL), and time-restricted high-fat diet (TRF) after initiation of the TRF protocol ($n = 8$ for NC, $n = 10$ for AL, $n = 10$ for TRF in **c**; $n = 7$ for NC, $n = 7$ for AL, $n = 8$ for TRF in **e**; *$p < 0.05$, **$p < 0.01$, ****$p < 0.0001$ by two-way ANOVA with Tukey's multiple comparisons test). The vertical bar indicates differences between NC and the AL and TRF groups. The horizontal bar indicates differences between AL and TRF at those time points. NC mice are colored blue, AL mice orange, and TRF mice green in all figure panels. **d, f** Daily mean food intake in OVX and VCD mice ($n = 8$ for NC, $n = 10$ for AL, $n = 10$ for TRF in **d**, $n = 8$ for NC, $n = 8$ for AL, $n = 8$ for TRF in **f**; **$p < 0.01$, ****$p < 0.0001$ by one-way ANOVA with Tukey's multiple comparisons test). **g, i** Intraperitoneal-glucose tolerance test (GTT) on female OVX or VCD mice after 7 weeks of TRF ($n = 7$ for NC, $n = 10$ for AL, $n = 9$ for TRF in **g**; $n = 7$ for NC, $n = 7$ for AL, $n = 8$ for TRF in **i**; *$p < 0.05$, **$p < 0.01$, ****$p < 0.0001$ by two-way ANOVA with Tukey's multiple comparisons test). The horizontal bar indicates differences between AL and TRF at those time points. **h, j** Fasting blood glucose (FBG) levels at the end of the study in OVX and VCD mice (**h**, $n = 8$ for NC, $n = 10$ for AL, $n = 10$ for TRF, in **j**; $n = 7$ for NC, $n = 7$ for AL, $n = 8$ for TRF; *$p < 0.05$, **$p < 0.01$, ****$p < 0.0001$ by one-way ANOVA with Tukey's multiple comparisons test). **k, m** Intraperitoneal-insulin tolerance test (ITT) in OVX or VCD mice after 8 weeks of TRF ($n = 8$ for NC, $n = 10$ for AL, $n = 10$ for TRF in **k**; $n = 7$ for NC, $n = 7$ for AL, $n = 8$ for TRF in **m**; *$p < 0.05$, **$p < 0.01$, ****$p < 0.0001$ by two-way ANOVA with Bonferroni's multiple comparisons test). **l, n** Fasting insulin levels at the end of the study in OVX and VCD mice ($n = 6$ for NC, $n = 7$ for AL, $n = 8$ for TRF in **l**; $n = 7$ for NC, $n = 7$ for AL, $n = 8$ for TRF in **n**; *$p < 0.05$, **$p < 0.01$, ***$p < 0.001$, ****$p < 0.0001$ by one-way ANOVA with Tukey's multiple comparisons test). For **b–n**, asterisks show statistical significance TRF vs AL or as shown. **o–q** Fasting glucose, ketone, and glucose–ketone index over 24 h in OVX mice $n = 8$ for AL, $n = 9$ for TRF. Period of fasting indicated by yellow bar, and period of feeding indicated by purple bar. Asterisks indicate statistical significance between AL and TRF (*$p < 0.05$, **$p < 0.01$, ****$p < 0.0001$), whereas dots indicate statistical significance for TRF vs 6 am (ZT0), (●$p < 0.05$, ●●$p < 0.01$, ●●●$p < 0.001$, ●●●●$p < 0.0001$) by two-way ANOVA with Bonferroni's multiple comparisons test. For **b–q**, all data are presented as mean ± SEM with $n$ representing the number of mice per group.

TRF improved insulin tolerance at 8 weeks (Fig. 1k, m, Supplementary Fig. 1f, h) and pyruvate tolerance, a measure of hepatic glucose production, at 9 weeks in both OVX and VCD models (Supplementary Fig. 1j–m). Fasting plasma insulin and homeostatic model of insulin resistance (HOMA-IR) were elevated in AL groups and significantly reduced by the TRF, indicating improved insulin sensitivity (Fig. 1l, n, Supplementary Fig. 1g, i). Glucose and ketone levels were constant in the AL group over 24 h in the OVX mouse model, but glucose levels fell while ketone levels rose over the fasting period in the TRF group. Glucose levels later rose and ketone levels fell with the onset of feeding in the TRF group (Fig. 1o, p). Consequently, the glucose–ketone index decreased over the period of fasting and increased with feeding in TRF mice (Fig. 1q).

Obesity is typically accompanied by hepatic steatosis. In the OVX model, TRF resulted in reduced hepatic lipid droplet number and size compared with AL mice (Supplementary Fig. 2a–c). At the gene expression level, we found that the lipid transport gene *Cd36* and the lipid storage genes *Cidea* and *Cidec* were elevated in the AL mice compared with NC, but were not elevated in the TRF group, consistent with the reduced steatosis (Supplementary Fig. 2d). Similar morphological changes were observed in the livers of VCD mice (Supplementary Fig. 2h–k). TRF also reduced the expression of several inflammatory genes relative to AL in the OVX mouse livers (Supplementary Fig. 2e, l). Phosphorylation of AKT and GSK3β was enhanced in livers from TRF mice compared with AL mice (Supplementary Fig. 2f, g). TRF mice exhibited reduced mammary and visceral fat mass compared with AL mice in both models, although the mice were still obese compared with NC mice (Supplementary Fig. 3a–d). Interestingly, in both of the mouse models TRF decreased the number of macrophage crown-like structures in the mammary fat pad (MFP) compared with AL mice (Supplementary Fig. 3e–g), and we observed a significant TRF effect on inflammatory gene expression in the MFP (Supplementary Fig. 3h, i). Thus, TRF stabilizes bodyweight without changing food intake, reduces hepatic steatosis, and improves glucose and insulin tolerance.

**TRF inhibits tumor growth in obese postmenopausal mice.** We then examined whether TRF reduced tumor growth in obese

OVX mice bearing orthotopic Py230 breast cancer cells (Supplementary Fig. 4a). Py230 cells form estrogen receptor and progesterone receptor-positive luminal-A type tumors when injected into the MFP[49]. TRF abrogated the tumor-promoting effects of obesity such that tumor growth in the TRF group was indistinguishable from that in the NC group and significantly less than the AL group (Fig. 2a, b, Supplementary Fig. 4b). The mean tumor weight in each mouse showed a positive correlation with fasting insulin levels and HOMA-IR (Fig. 2c and Supplementary Fig. 4c). The effect of TRF on Py230 tumor growth was confirmed in the obese VCD model (Fig. 2d, e, Supplementary Fig. 4d). As seen for the OVX model, the tumor weight correlated with fasting insulin and HOMA-IR in VCD mice (Fig. 2f and Supplementary Fig. 4e). To test whether this TRF effect on tumor growth was also observed in lean mice, we performed similar TRF intervention experiments in lean mice. TRF of NC-fed mice (NC-TRF) resulted in no significant change in bodyweight, mammary fat and visceral fat weights, or food intake compared to NC ad libitum group (NC-AL) (Supplementary Fig. 4f–j). We also found no difference in tumor growth between NC-AL mice and NC-TRF mice (Supplementary Fig. 4k–m). These results indicate that TRF was equally effective at inhibiting tumor growth in two different models of postmenopausal obesity.

As accelerated tumor growth is associated with enhanced proliferation and increased vascularization[50], we assessed the effect of TRF on these parameters in tumors from OVX mice. We observed a reduction in mitotic events and Ki67-positive nuclei in tumor sections of TRF mice compared to AL mice (Fig. 2g–i). Tumors from AL mice showed enhanced CD31 endothelial staining compared with NC mice, and TRF reduced tumor vascularization to levels found in NC mice (Fig. 2g, j). Further, phosphorylation of AKT and GSK3β was higher in AL tumors compared with TRF tumor tissue in the OVX mice, suggesting that TRF is able to attenuate tumor growth by modulating the AKT pathway (Fig. 2k–m). Phosphorylation of ERK1/2 was not affected by either AL or TRF diets (Fig. 2k, n).

To determine whether we could replicate the beneficial effect of TRF on tumor growth using a more aggressive triple-negative breast cancer cell line (ER/PR/HER2 negative), we injected obese OVX mice with E0771 cells (Supplementary Fig. 5a). In concordance with

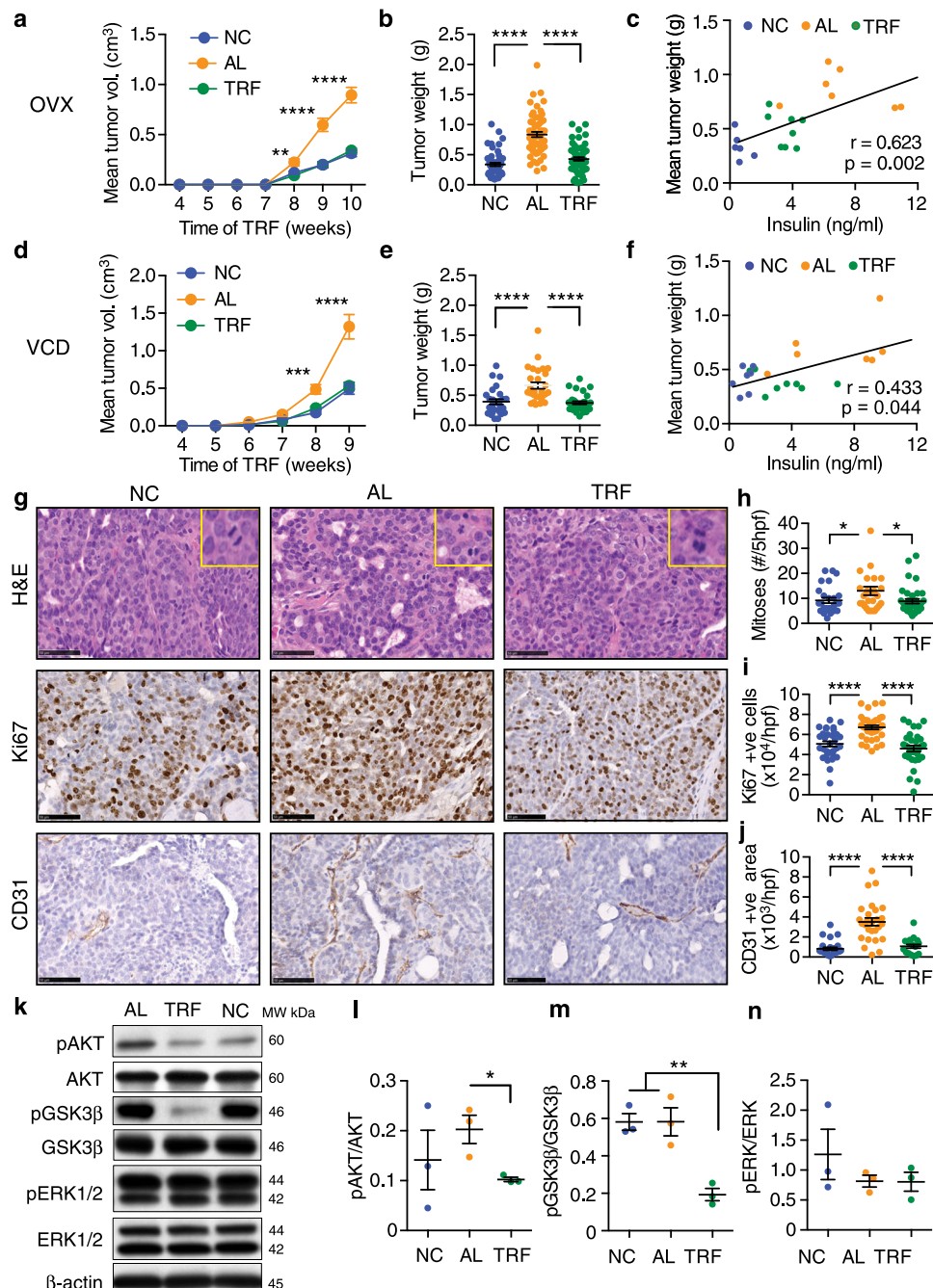

**Fig. 2 TRF reduces Py230 mammary tumor growth in two obese postmenopausal mouse models. a**, **d** Py230 tumor volume over time in OVX or VCD mice on NC, AL, and TRF (number of mice $n = 8$ for NC, $n = 10$ for AL, $n = 10$ for TRF in **a**; $n = 7$ for NC, $n = 7$ for AL, $n = 7$ for TRF in **d**; **$p < 0.01$, ***$p < 0.001$, ****$p < 0.0001$ by two-way ANOVA with Tukey's multiple comparisons test). **b**, **e** Individual tumor weights at the end of the study in OVX or VCD mice (number of tumors $n = 32$ for NC, $n = 40$ for AL, $n = 40$ for TRF in **a**; $n = 27$ for NC, $n = 28$ for AL, $n = 32$ for TRF in **d**; ****$p < 0.0001$ by one-way ANOVA with Tukey's multiple comparisons test). **c**, **f** Correlation plot of mean tumor weight per mouse vs. fasting insulin in OVX or VCD mice (for **c**, number of mice $n = 6$ for NC, $n = 7$ for AL, $n = 8$ for TRF; for **f**, $n = 7$ for NC, $n = 7$ for AL, $n = 8$ for TRF). Data were analyzed by linear regression using Pearson correlation. **g** Representative H&E, Ki67 and CD31 staining of Py230 tumor tissue in OVX mice on NC, AL, and TRF. Scale bar, 50 μm. Insets in H&E images show higher magnification of a mitotic cell. Data shown are representative of two independent experiments. **h**–**j** Quantification of mitoses ($n = 27$ for NC, $n = 38$ for AL, $n = 36$ for TRF), Ki67-positive cells ($n = 31$ for NC, $n = 36$ for AL, $n = 32$ for TRF), and CD31 staining ($n = 27$ for NC, $n = 27$ for AL, $n = 21$ for TRF). Data are presented as mean ± SEM per high-powered field (hpf) or 5hpf for mitoses, with 4 hpf per section ($n =$ number of tumors). Asterisks indicate significance; *$p < 0.05$, ****$p < 0.0001$ by one-way ANOVA with Tukey's multiple comparisons test. **k** Akt(Ser473), GSK3β(Ser9), and ERK(Thr202/Tyr204) phosphorylation by western blot in Py230 tumor extracts. **l**–**n** Quantification of Akt, GSK3β, and ERK phosphorylation normalized to total protein (mean ± SEM, $n = 3$, two-tailed $t$ test). Data are presented as mean ± SEM. Asterisks show statistical significance TRF vs AL or as shown.

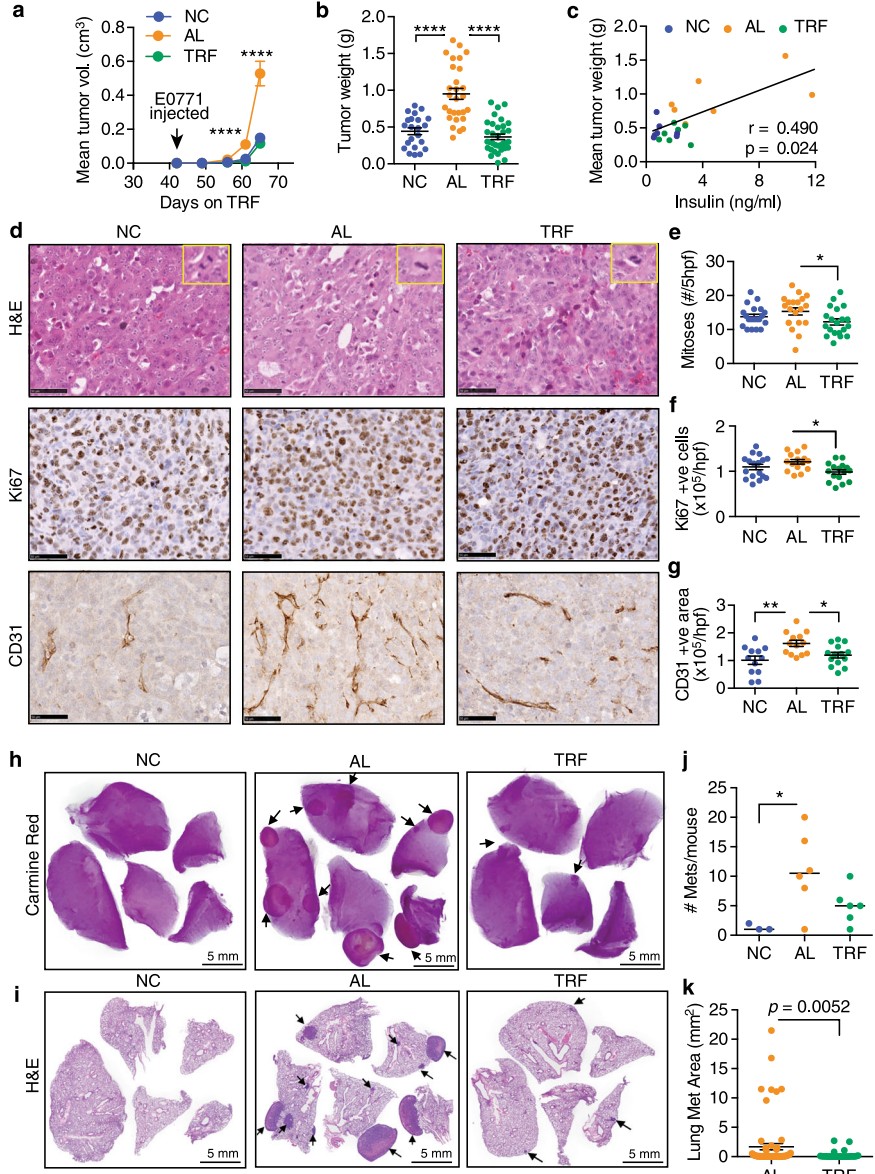

**Fig. 3 TRF attenuates E0771 mammary tumor growth and metastasis to lung in obese OVX mice. a** E0771 tumor volume over time in OVX mice on NC, AL, and TRF (number of mice $n = 7$ for NC, $n = 7$ for AL, $n = 9$ for TRF; ****$p < 0.0001$ by two-way ANOVA with Tukey's multiple comparisons test). **b** Individual tumor weight at sacrifice (number of tumors $n = 22$ for NC, $n = 28$ for AL, $n = 34$ for TRF). ****$p < 0.0001$ by one-way ANOVA with Tukey's multiple comparisons test. **c** Correlation plot of mean tumor weight per mouse vs. fasting insulin (number of mice $n = 7$ for NC, $n = 7$ for AL, $n = 7$ for TRF). Data were analyzed by linear regression using Pearson correlation. **d** Representative H&E, Ki67, and CD31 staining of E0771 tumor tissue in OVX mice on NC, AL, and TRF. Scale bar, 50 μm. Insets in H&E images show higher magnification of a mitotic cell. Data shown are representative of two independent experiments. **e–g** Quantification of mitoses ($n = 20$ sections for all groups), Ki67-positive cells ($n = 17$ for NC, $n = 15$ for AL, $n = 16$ for TRF), and CD31-positive cells ($n = 12$ for NC, $n = 13$ for AL, $n = 14$ for TRF) in sections of E0771 tumor tissue. Data are presented as mean ± SEM per high-powered field (hpf) or 5 hpf for mitoses, with 4 hpf per section ($n = $ number of tumors). Asterisks indicate significance; *$p < 0.05$, **$p < 0.01$ by one-way ANOVA with Tukey's multiple comparisons test. **h** Carmine Red staining of whole-mount lungs showing macro-metastases of E0771 tumor cells. **i** H&E stained lung sections showing metastases of E0771 tumor cells. Data shown are representative of two independent experiments. **j** Total number of metastases per mouse (number of mice $n = 3$ for NC, $n = 6$ for AL, $n = 6$ for TRF; *$p < 0.05$ by one-way ANOVA with Tukey's multiple comparisons test). **k** Cross-sectional area of individual lung metastases (number of mice $n = 3$ for NC, $n = 6$ for AL, $n = 6$ for TRF; two-tailed $t$ test). Data are presented as mean ± SEM. For **a**, **c**, **h**, and **j**, $n$ represents number of mice per group. Asterisks show statistical significance TRF vs AL or as shown.

our previous studies, we observed that obesity-enhanced the growth of E0771 tumors in the AL group, and that TRF completely eliminated the accelerated growth (Fig. 3a, b, and Supplementary Fig. 5b), indicating that TRF was also highly effective in reducing the growth of a more aggressive breast cancer subtype. The mean tumor weight in each mouse showed a positive correlation with fasting insulin levels and HOMA-IR (Fig. 3c and Supplementary

Fig. 5c). As before, we assessed the effect of TRF on proliferation and vascularization in the E0771 tumors. TRF reduced tumor cell mitoses and Ki67 staining, and in addition reduced the level of CD31-positive tumor vascularization to levels found in NC mice (Fig. 3d, g). Since obesity is also a risk factor for distant metastasis and E0771 cells are highly metastatic, we injected E0771 cells via the tail vein into obese OVX mice and assessed lung metastasis. Lung

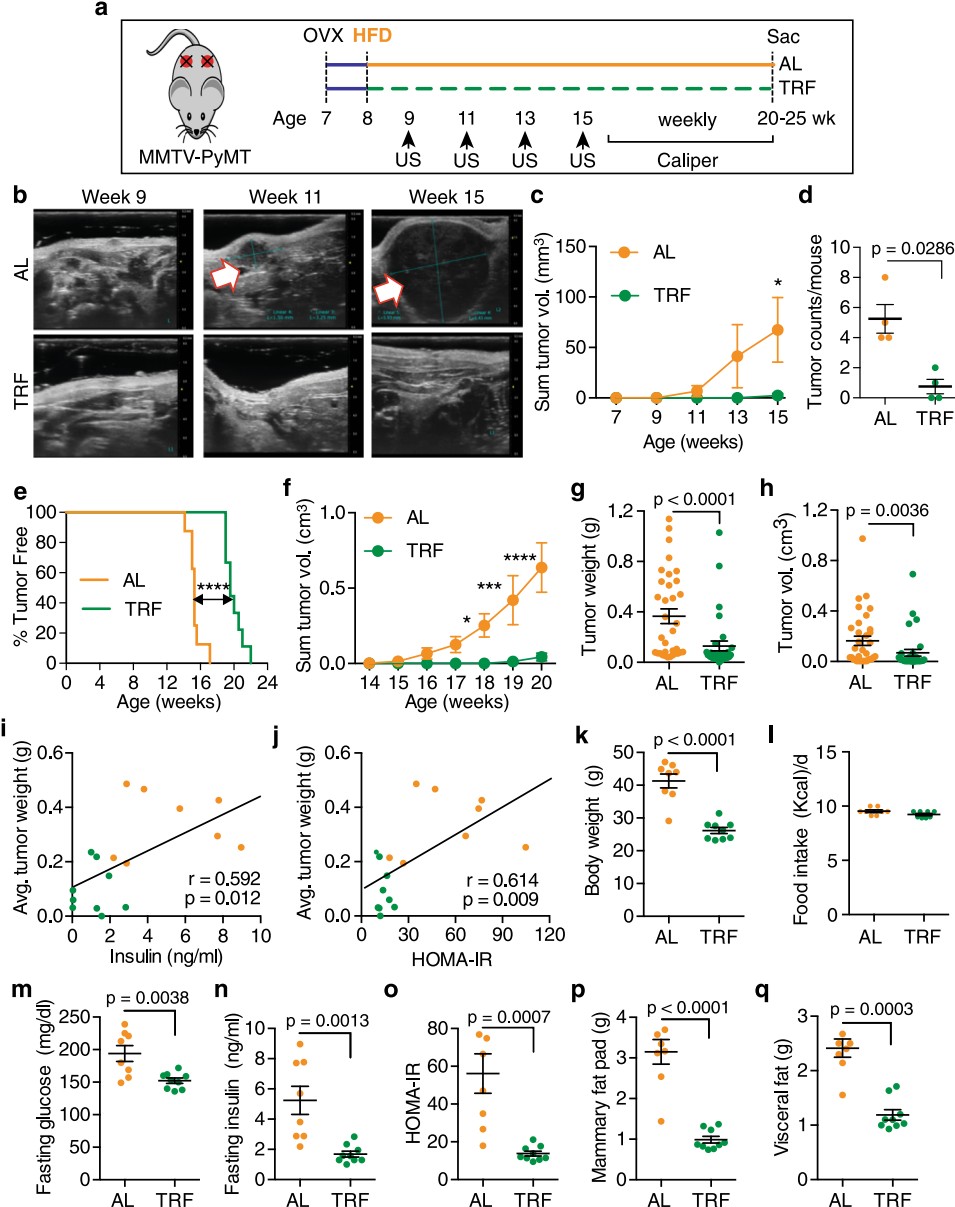

**Fig. 4 TRF inhibits tumor initiation and growth in obese OVX-PyMT transgenic mice. a** Schematic showing the experimental design. **b** Representative ultrasound images of mammary fat pads from mice on AL or TRF at post-natal weeks 9, 11, and 15. White arrow indicates a tumor. Data shown are representative of two independent experiments. **c** Tumor volume measured weekly by ultrasound (number of mice $n = 4$, *$p < 0.05$ by two-way ANOVA with Bonferroni's multiple comparisons test). **d** Number of tumors per mouse at week 15 (number of mice $n = 4$; two-tailed $t$ test). **e** Tumor-free survival plot for mice on AL or TRF. Survival analysis was performed by Log-rank (Mantel–Cox) test, ****$p < 0.0001$, number of mice $n = 8$ for AL and $n = 9$ for TRF group. **f** Total tumor volume per mouse measured weekly with calipers (number of mice $n = 8$ for AL and $n = 9$ for TRF; *$p < 0.05$, ***$p < 0.001$, ****$p < 0.0001$ by 2-way ANOVA with Bonferroni's multiple comparisons test). **g, h** Individual tumor weight and volume at the end of the study (number of tumors $n = 34$ for AL and $n = 32$ for TRF; two-tailed $t$ test). **i, j** Plots showing correlation of mean tumor weight per mouse vs. fasting insulin level or HOMA-IR (number of mice $n = 8$ for AL and $n = 9$ for TRF). Data were analyzed by linear regression using Pearson correlation. **k** Body weight at end of study (number of mice $n = 8$ for AL, $n = 9$ for TRF; ****$p < 0.0001$ by two-tailed $t$ test). **l** Caloric intake ($n = 8$), **m–o** Fasting blood glucose, insulin, and HOMA-IR at end of study (number of mice $n = 8$ for AL, $n = 9$ for TRF; two-tailed $t$ test). **p, q** Mammary and visceral fat tissue weight at end of study (number of mice $n = 8$ for AL, $n = 9$ for TRF; two-tailed $t$ test). Data presented as mean ± SEM. Asterisks show statistical significance between AL and TRF. AL mice are labeled in orange, and TRF mice in green.

sections demonstrated a significant reduction of macro-metastases following TRF compared with AL mice (Fig. 3h, k).

We then assessed the effect of TRF on breast tumor initiation and growth in the transgenic MMTV-polyoma middle T antigen (PyMT) model of mammary tumorigenesis in the C57Bl/6 background (Fig. 4a). As the tumor latency in the transgenic PyMT model is 14 weeks, we initiated TRF and HFD feeding concurrently when the mice were 8 weeks of age, 1 week after

OVX. Tumor initiation was imaged using high-frequency ultrasonography of the MFPs (Fig. 4b). TRF caused a significant 4–5 week delay in tumor initiation (Fig. 4b–d). Once tumors were palpable, tumor growth was followed by caliper measurements. TRF delayed palpable tumor appearance and growth (Fig. 4e, f). At the end of the study, TRF tumor volume and weight were significantly lower in comparison with the AL group (Fig. 4g, h). As in the orthotopic cancer models, tumor weight correlated with

fasting insulin and HOMA-IR (Fig. 4i, j). At the end of the study, the AL animals were obese (>40 g), whereas the TRF group remained lean (<30 g) (Fig. 4k). Importantly, as in the other models, the TRF group consumed the same number of calories as the AL group (Fig. 4l). The fasting blood glucose, insulin, and HOMA-IR were all significantly lower in the TRF group compared with the AL group (Fig. 4m–o). Consistent with the bodyweight, the MFP and visceral fat pad weights were also decreased significantly in TRF mice (Fig. 4p, q).

To test the effect of TRF on pre-existing tumors, we placed 9-week old OVX C57BL/6 J mice on HFD for 10 weeks to make them obese, then injected Py230 cells into the MFPs (Fig. 5a). Three weeks later MFPs were imaged by ultrasound to verify the

presence of tumors. Then, mice were randomized into AL, TRF, or NC groups as previously described. Tumor growth was measured by calipers once tumors were palpable. All tumors continued to grow but the tumors in the AL grew much larger than in the TRF or NC groups and the tumors in the TRF group grew to a similar size as tumors in the NC group (Fig. 5b). At the end of the study at 25 weeks of age, the tumors were larger and heavier in the AL group compared with the TRF and NC groups (Fig. 5c, d). TRF resulted in improved glucose tolerance compared with the AL group (Fig. 5e) and fasting glucose was reduced (Fig. 5f) confirming the improved metabolic status of the mice on TRF. TRF mice showed reductions in bodyweight, liver weight, and MFP but not visceral fat pad weight compared

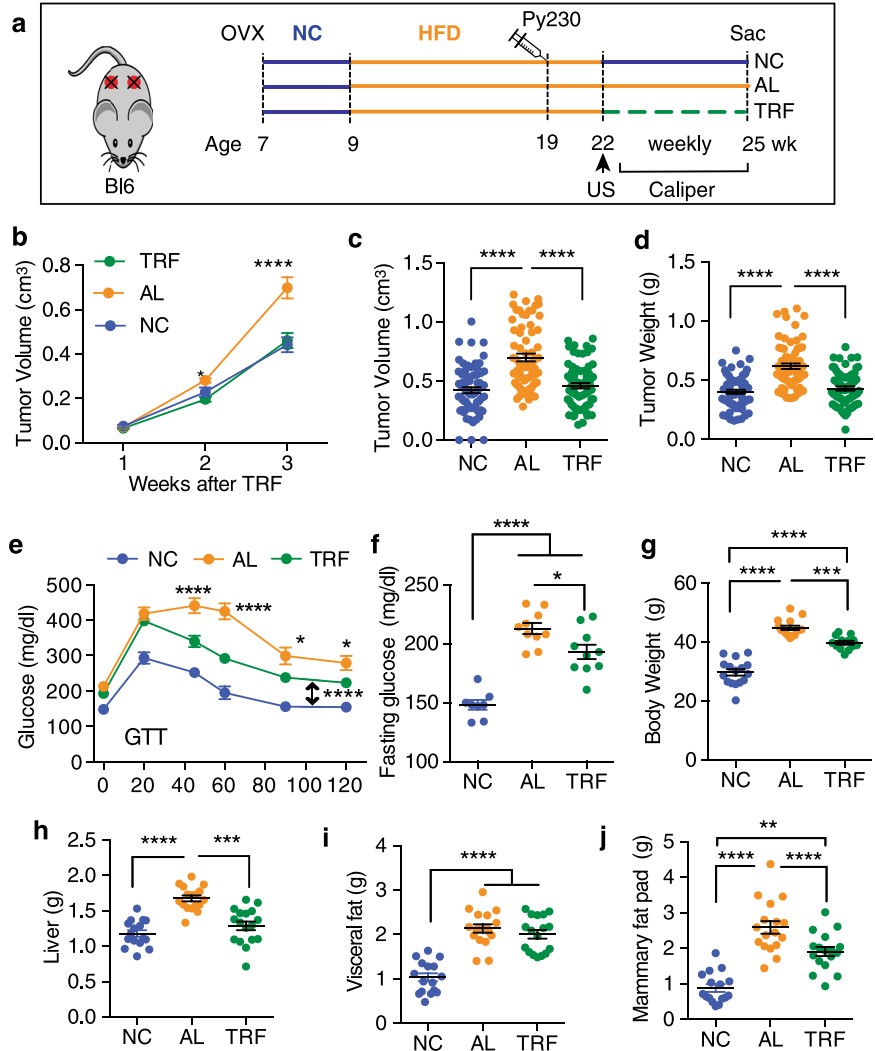

**Fig. 5 TRF inhibited growth of pre-existing Py230 mammary tumors in obese postmenopausal mouse model. a** Schematic showing experimental design with Py230 breast tumor injection 3 weeks prior to initiation of TRF. OVX mice are placed on HFD for 10 weeks to become obese then injected with py230 cells in the mammary fat pad. Three weeks after tumor cells injection the tumors were measure by ultrasonography imaging system (US) to confirm tumor development and randomized into AL or TRF HFD groups or NC group for a further 3 weeks and tumor size was measured weekly with calipers. **b** Py230 tumor volume over time following TRF (number of mice $n = 17$, $*p < 0.05$, $****p < 0.0001$ by two-way ANOVA with Tukey's multiple comparisons test). **c**, **d** Tumor volume and weight at the end of the study (number of tumors $n = 67$ for NC, $n = 86$ for AL and TRF, $****p < 0.0001$ by two-way ANOVA with Tukey's multiple comparisons test, $n$ represents number of tumor). **e** Intraperitoneal-glucose tolerance test (GTT) after 3weeks of TRF (number of mice $n = 8$ for NC, $n = 10$ each for AL and TRF, $*p < 0.05$, $****p < 0.0001$ by two-way ANOVA with Tukey's multiple comparisons test). **f** Fasting blood glucose at end of study (number of mice $n = 8$ for NC, $n = 10$ each for AL and TRF, $*p < 0.05$, $****p < 0.0001$ by 1-way ANOVA with Tukey's multiple comparisons test). **g** Bodyweight at end of study (number of mice $n = 16$ for NC, $n = 15$ for AL, $n = 13$ for TRF, $***p < 0.001$, $****p < 0.0001$ by one-way ANOVA with Tukey's multiple comparisons test). **h**- **j** Liver tissue, visceral fat, and mammary fat weight at end of study (number of mice $n = 16$ for NC, $n = 17$ for AL, $n = 17$ for TRF, $**p < 0.01$, $***p < 0.001$, $****p < 0.0001$ by one-way ANOVA with Tukey's multiple comparisons test). Data presented as mean ± SEM. Asterisks indicate statistical significance as shown.

to the AL group (Fig. 5g, j), further corroborating metabolic improvement.

**Insulin is a mitogen for Py230 breast cancer cells in vitro**. We next tested whether insulin was acting directly on the Py230 tumor cells to stimulate proliferation. Insulin caused a dose-dependent increase in cell number (Fig. 6a) and a dose-dependent increase in Py230 cell migration in a transwell assay (Fig. 6b, c). Similar results were observed in a scratch wound closure assay (Fig. 6d, e). We also carried out tumor spheroid formation assays following insulin treatment. The cells formed more spheroids in response to insulin, and interestingly formed two types of spheroids, round, and irregular (Fig. 6f–j). The proportion of irregularly shaped spheroids increased following insulin treatment compared with untreated cells (Fig. 6j). We

also tested whether the elevated ketones observed during the fasting could affect breast cancer cell survival in vitro. The addition of the ketone body β-hydroxybutyrate did not inhibit Py230 or E0771 viability in the presence or absence of serum, nor could it support cell survival in the absence of glucose (Supplementary Fig. 6a–d). Thus, insulin promoted breast cancer cell proliferation, migration, and tumor formation in vitro. We also tested whether estrogen and progesterone could stimulate Py230 cell proliferation in vitro. Both estradiol and progesterone were able to stimulate Py230 cell proliferation independently of insulin (Supplementary Fig. 6e, f) suggesting the presence of estrogen and progesterone receptors in the cells. Indeed, staining of Py230 tumors from the mice demonstrated that they were ER and PR positive (Supplementary Fig. 6g, h).

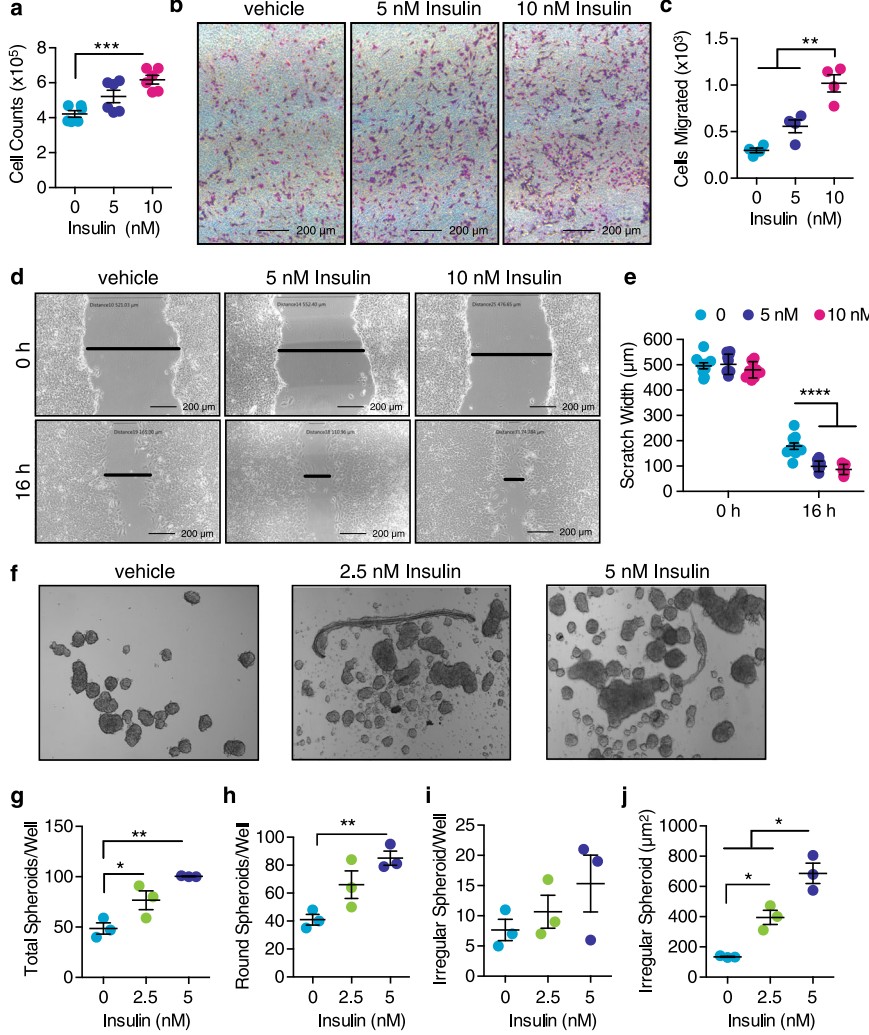

**Fig. 6 Insulin stimulates Py230 cell proliferation, motility, migration, and tumor spheroid formation. a** Cell counts of Py230 breast cancer cells after treatment with 0, 5, or 10 nM insulin for 24 h ($n = 6$ replicates of treatment per group, ***$p < 0.01$ by one-way ANOVA with Tukey's multiple comparisons test). Vehicle treated cells are colored cyan, cells treated with 5 nM insulin in blue, and cells treated with 10 nM insulin in magenta. **b** Representative image of Py230 cell migration in transwell chambers following 0, 5, or 10 nM insulin treatment for 24 h. **c** Quantitation of cells that have migrated through transwell membrane ($n = 4$ replicates of treatment per group performed twice, **$p < 0.01$ by one-way ANOVA with Tukey's multiple comparisons test). **d** Representative images of Py230 cell migration into a scratch wound after 0, 5, or 10 nM insulin treatment for 16 h. Black bars indicate distance between sides of scratch. **e** Quantitation of scratch width ($n = 10$, $n$ represents number of scratches from triplicate experiment performed twice, ****$p < 0.0001$ by two-way ANOVA with Tukey's multiple comparisons test). **f** Representative image of tumor spheroids formed by Py230 cells in response to 2.5 or 5 nM insulin for 7 days. **g–j** Quantification of total number of tumor spheroids, number of round spheroids, number of irregular spheroids, and area of irregular spheroids per well ($n = 3$ replicates of treatment per group performed twice, *$p < 0.05$, **$p < 0.01$ by one-way ANOVA with Tukey's multiple comparisons test). Vehicle treated cells are colored cyan, cells treated with 2.5 nM insulin in green, and cells treated with 5 nM insulin in blue. Data are presented as mean ± SEM. Asterisks show statistical significance as indicated.

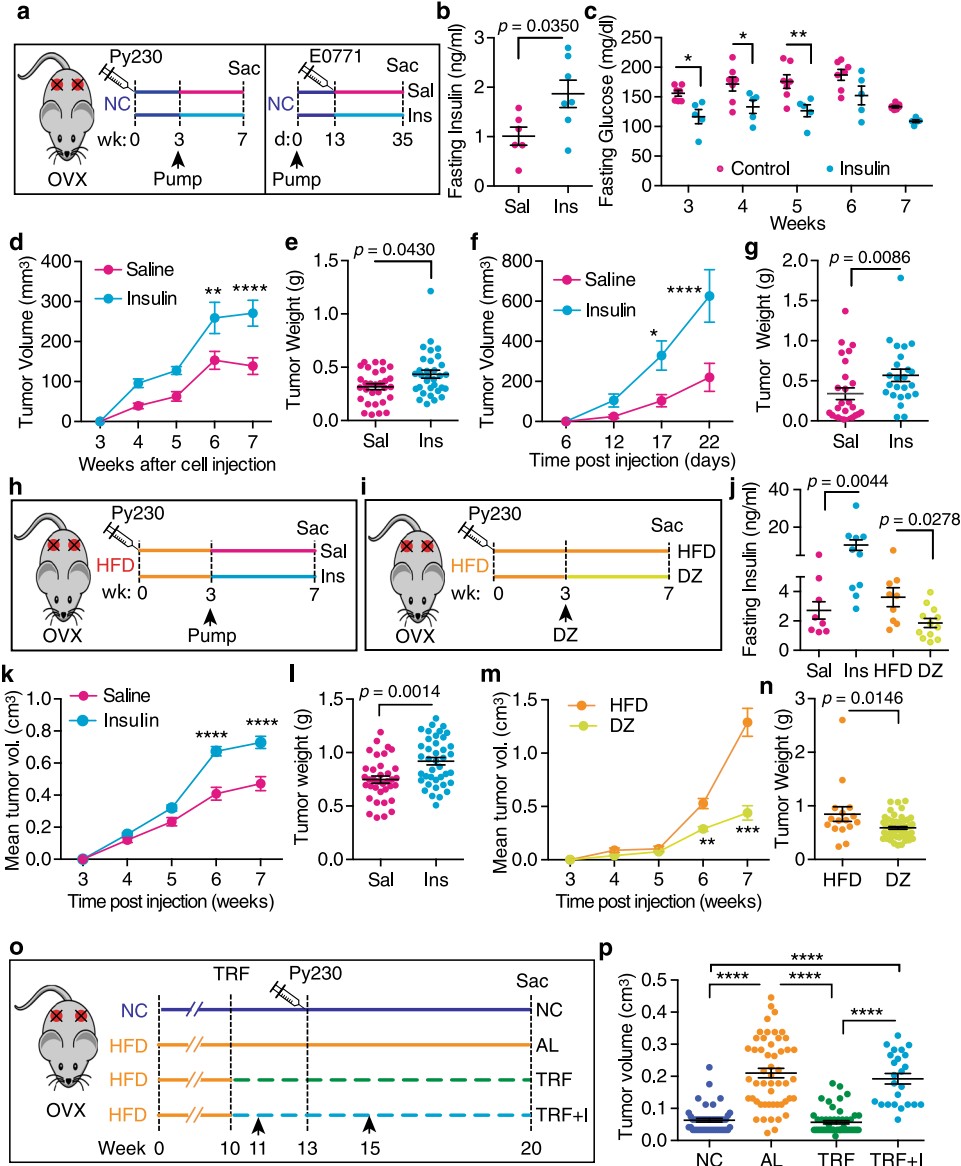

**Hyperinsulinemia drives tumor growth**. To test whether hyperinsulinemia alone augments tumor growth in postmenopausal mice, we delivered exogenous insulin to lean mice by implanting osmotic pumps (Fig. 7a). A dose of 0.2 U/day insulin increased circulating insulin levels approximately twofold and reduced glucose levels 10–20% without generating hypoglycemia (Fig. 7b, c). Growth of Py230 tumors was accelerated in mice with insulin pumps compared to mice with saline pumps (Fig. 7d, e). A similar acceleration was seen in the more aggressive E0771 tumors (Fig. 7f, g), indicating that hyperinsulinemia alone can drive tumor growth in our breast cancer models. We also evaluated the effect of insulin in augmenting obesity-driven Py230 tumor growth by implanting insulin pumps in HFD-fed mice (Fig. 7h). These pumps delivered 0.6 U/day insulin as the obese mice are insulin resistant. To test sufficiency, we reduced circulating insulin levels in HFD-fed mice using the K-ATP channel agonist diazoxide (DZ) (Fig. 7i). The resulting insulin levels were significantly greater in the insulin pump-bearing HFD-fed mice and reduced in the DZ mice compared with the HFD-fed controls (Fig. 7j). The insulin pump-bearing HFD-fed mice had significantly lower glucose compared with the HFD-fed controls but did not become hypoglycemic (Supplementary Fig. 7). Confirming our insulin

results in the NC-fed mice, insulin increased tumor growth in HFD mice compared to mice with saline pumps (Fig. 7k, l). Importantly, tumor growth was inhibited in the DZ group in comparison with the HFD control group (Fig. 7m, n). At last, we tested whether artificial hyperinsulinemia would overcome the beneficial effect of TRF on tumor growth using the Py230 orthotopic model (Fig. 7o). Elevating insulin using a mini-pump completely abrogated the effect of TRF on inhibiting tumor growth (Fig. 7p). Together, these results demonstrate that obesity-induced hyperinsulinemia is the driver of mammary tumor growth in these obesity models and that TRF can reduce tumor growth by reducing hyperinsulinemia.

**Tumor circadian rhythms are dysregulated in obesity but restored by TRF.** TRF is known to synchronize liver circadian rhythms dysregulated by obesogenic diets[19,47], however the effect of TRF on modulating circadian rhythms in MFPs or tumors has not been investigated. Therefore, we examined the expression of core clock genes in liver, MFP, and tumor tissue from the NC, AL, TRF groups of mice harvested every 4 h over 24 h (ZT0, 4, 8, 12, 16, 20). Initially, we examined clock gene expression in mice

**Fig. 7 Hyperinsulinemia drives Py230 tumor growth in obese OVX mice. a** Schematic showing design of insulin pump study in NC-fed OVX mice. Py230 cells are orthotopically injected at week 0 then insulin pumps delivering 0.2 U/day are implanted at week 3 and tumors followed to week 7. For the E0771 cells, insulin pumps delivering 0.2 U/day are implanted at day 0 then E0771 cells orthotopically injected on day 13 and tumors followed to day 35. Mice with saline pumps (Sal) are colored magenta and mice with insulin pumps (Ins) are labeled cyan. **b** Fasting plasma insulin at end of study (number of mice $n = 6$ for Sal, $n = 7$ for Ins; two-tailed $t$ test). Mice with saline pumps are shown in magenta, mice with insulin pumps in cyan. **c** Fasting blood glucose measured weekly in mice on NC following insulin pump implantation (number of mice $n = 7$ for control, $n = 5$ for insulin; *$p < 0.05$, **$p < 0.01$ by two-way ANOVA with Bonferroni's multiple comparisons test). **d** Mean Py230 tumor volume measured weekly with calipers after insulin pump implantation (number of mice $n = 8$ per group; **$p < 0.01$, ****$p < 0.0001$ by two-way ANOVA with Bonferroni's multiple comparisons test). **e** Mean individual Py230 tumor weight at end of study (number of tumors $n = 30$ for Sal, $n = 32$ for Ins; two-tailed $t$ test). **f** Mean E0771 tumor volume measured weekly after insulin pump implantation (number of mice $n = 10$ for saline, $n = 8$ for insulin; *$p < 0.05$, ****$p < 0.0001$ by two-way ANOVA with Bonferroni's multiple comparisons test). **g** Mean individual E0771 tumor weight at end of study (number of tumors $n = 27$ for Sal, $n = 24$ for Ins; two-tailed $t$ test). **h** Schematic showing design of insulin pump study in obese OVX mice on HFD. Py230 cells are orthotopically injected at week 0 then pumps delivering 0.6 U/day insulin (Ins, cyan) or saline (Sal, magenta) are implanted at week 3. **i** Schematic showing design of diazoxide study in obese OVX mice on HFD. Py230 cells are orthotopically injected at week 0, mice are switched to HFD (orange) or HFD + diazoxide diet (DZ, yellow) at week 3, then tumors are followed to week 7. **j** Fasting plasma insulin at end of study (number of mice $n = 8$ for Sal, $n = 10$ for Ins, $n = 9$ for HFD, $n = 12$ for DZ, one-way ANOVA with Tukey multiple comparison test). **k** Mean Py230 tumor volume measured weekly in insulin pump groups (number of mice $n = 9$ for Saline, $n = 10$ for Insulin; ****$p < 0.0001$ by two-way ANOVA with Bonferroni's multiple comparisons test). **l** Individual Py230 tumor weight at the end of the pump study (number of tumors $n = 36$ for Sal, $n = 40$ for Ins; **$p(0.0014)$ by two-tailed $t$ test). **m** Mean Py230 tumor volume measured weekly in DZ mice group (number of mice $n = 4$ for HFD, $n = 14$ for DZ; **$p < 0.01$, ****$p < 0.0001$ by two-way ANOVA with Bonferroni's multiple comparisons test). **n** Individual Py230 tumor weight at the end of the diazoxide study (number of tumors $n = 16$ for HFD, $n = 55$ for DZ; two-tailed $t$ test). **o** Schematic of the TRF study with implanted insulin pumps in obese OVX mice. Arrowheads indicate implantation of pumps on weeks 11 and 15. Py230 cells injected at week 13. Mice on NC are shown in blue, mice on AL in orange, mice on TRF in green and mice on TRF with insulin pumps (TRF + I) in cyan. **p** Py230 individual tumor volume after 10 weeks (number of tumors $n = 48$ for NC, $n = 52$ for AL and TRF, $n = 24$ for TRF + I; ****$p(0.0001)$ by one-way ANOVA with Tukey's multiple comparisons test). Data presented as mean ± SEM. Asterisks show statistical significance as indicated.

fed NC to determine whether the MFP and tumors have circadian rhythms. As expected, the liver showed robust circadian rhythms for the *Bmal1, Per1, Per2, Cry1,* and *Reverba* genes, but *Clock, Rora,* and *Cry2* genes were acyclic (Fig. 8a). The circadian parameters such as $p$ value and phase by RAIN analysis, and amplitude, phaseshift and peak ZT by curve fitting are given in Supplementary Table 1. The MFP also showed robust circadian rhythms for the *Bmal1, Per2, Cry1*, and *Reverba* genes, but *Clock, Cry2, Per1*, and *Rora* were acyclic (Fig. 8a). Although the amplitudes of the oscillations were different between liver and MFP, the phase of the rhythms was the same. In contrast in the tumors, *Bmal1, Clock, Per1, Per2, Cry2, Rora*, and *Reverba* showed circadian rhythms, but *Cry1* was acyclic (Fig. 8a). Surprisingly, the circadian rhythms in *Per1, Per2, Clock, Cry1,* and *Reverba* expression showed the same phase with maximal expression around ZT4, but *Bmal1* and *Rora* showed a 4 h phase advance with peak expression around ZT0. Only the phases of the *Reverba* and *Bmal1* genes were not different between the three tissue types (Fig. 8a). We then investigated the changes in each gene in each tissue following the three dietary conditions. In liver, *Bmal1* showed circadian rhythms under all three conditions, but the amplitude was lowest in the AL group (Fig. 8b). The circadian expression of *Per1* and *Cry1* was suppressed in the AL group compared with the NC group and *Cry1* rhythm was restored in the TRF group (Fig. 8b). In the MFP, rhythm of *Bmal1* was suppressed in the AL group compared with the NC group and restored in the TRF group; *Per1* was cyclic in the NC group, but acyclic in the AL and TRF groups; and *Cry1* was partially suppressed in the AL group compared with the NC group and restored in the TRF group (Fig. 8c). In the tumors, basal *Bmal1, Per1,* and *Cry1* expression was increased in both groups fed HFD compared with NC (Fig. 8d). The *Bmal1* gene showed a ~6 h phase delay in the AL group compared with the NC group that was corrected by TRF. Circadian rhythm of the *Per1* gene was suppressed in the AL group compared with the NC group and cyclicity was restored by TRF. The *Cry1* gene was acyclic in tumors of mice in all groups (Fig. 8d), whereas its expression is cyclic in liver and mammary fat. *Clock, Per2, Cry2, Rora,* and *Reverba* also showed differences in circadian gene expression in

the three tissues between the NC, AL and TRF groups (Supplementary Fig. 8). Notably, in tumors the circadian rhythms of the *Clock, Per2, Cry2,* and *Rora* genes were suppressed in the AL group compared with NC group and restored by TRF, as was the *Per1* gene. Circadian differences in gene expression were confirmed at the protein level in the tumors (Fig. 9) and in the liver and MFP (Supplementary Figs. 9 and 10). BMAL1 Ser42 phosphorylation showed circadian pattern differences between the groups in the tumors, with highest expression around ZT4-8, and highest level in the AL group compared with the NC and TRF groups (Fig. 9a, b). BMAL1 protein was acyclic in the NC group but circadian in the AL and TRF groups. PER1 levels were cyclic in all groups but showed more robust oscillation in the TRF group (Fig. 9e, f) with maximal expression at ~ZT10; CRY1 levels, in contrast, were only circadian in the TRF group (Fig. 9c, d) with peak expression at ~ZT4. The circadian parameters are given in Supplementary Table 2. As expected, peaks in protein expression occurred after peaks in mRNA expression (Fig. 8). TRF enhanced the circadian rhythms of BMAL1 phosphorylation, and PER1 and CRY1 proteins in the liver compared to the NC and AL group (Supplementary Fig. 9). In the MFP BMAL1 and CRY1 proteins showed circadian differences by group but PER1 was not detected (Supplementary Fig. 10). BMAL1 Ser42 phosphorylation was cyclic only in the TRF group in the MFP and the phase of CRY1 protein oscillations was shifted 10 h by TRF.

The data above suggested that the tumor cells have a circadian clock, so we tested whether the Py230 cells have circadian rhythms in vitro. Serum-starved Py230 cells were shocked with horse serum in the presence and absence of exogenous insulin and gene expression measured over 48 h. *Bmal1, Per1,* and *Cry1* all showed cyclical changes in expression (Fig. 8e). Interestingly, only *Per1* showed a circadian rhythm (24 h). The *Bmal1* gene showed a cycling period of 40 h, and *Cry1* a cycling period of 32 h. *Per2, Cry2, Rora,* and *Reverba* also showed cyclical change of 24 h (Supplementary Fig. 11a). Insulin had little effect in the context of the serum shock (Fig. 8e), so we tested whether the genes were insulin-responsive in vitro. Insulin-stimulated expression of circadian genes in both Py230 cells and E0771 cells (Supplementary Fig. 11b, c). Overall, whereas the liver and MFP

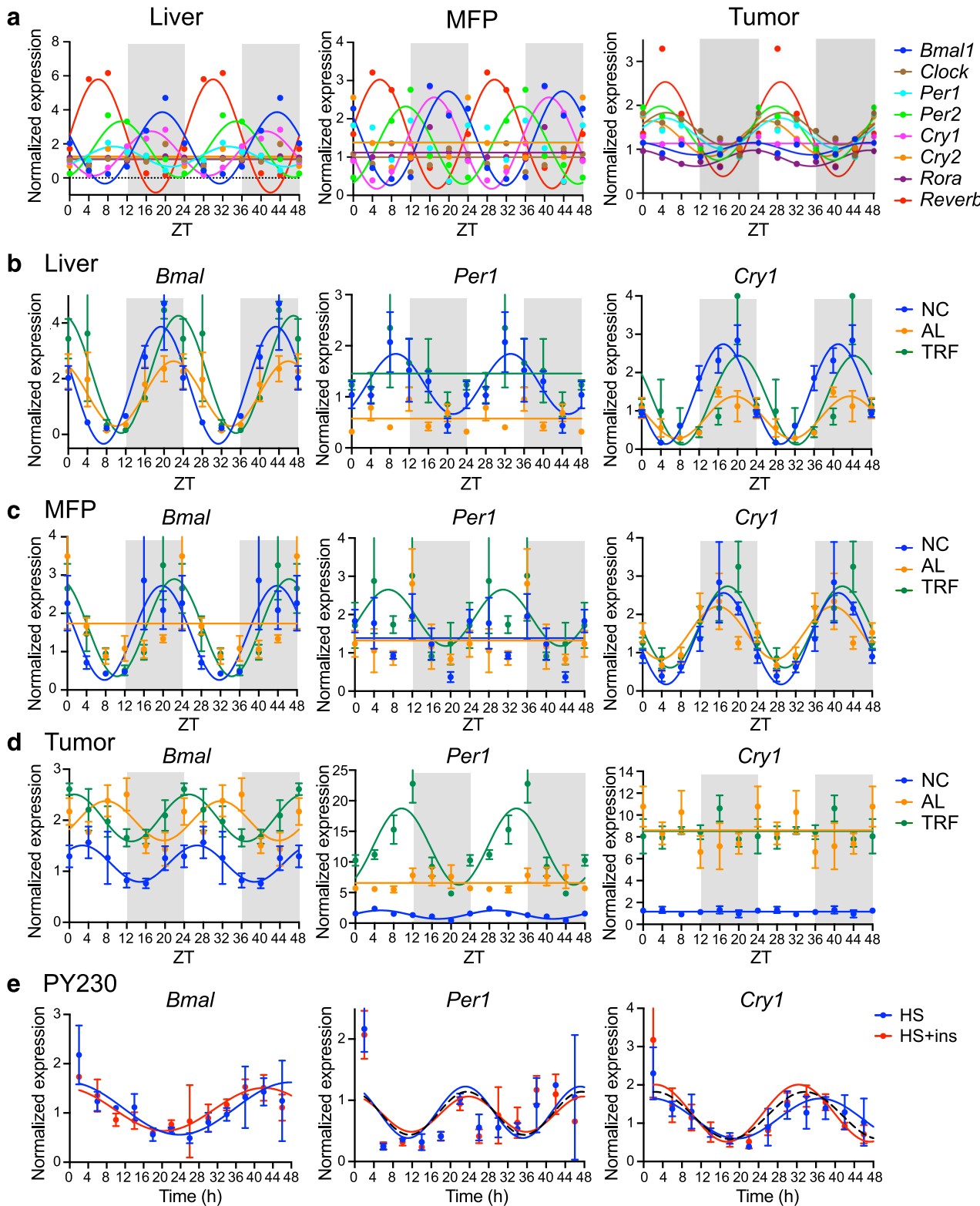

showed nutritional and timing-sensitive circadian rhythms with the expected phase relationships, the tumor had disrupted circadian rhythms phases that were still sensitive to nutritional status and timing.

## Discussion

Although it is often presumed that a healthy diet will improve breast cancer outcomes, data from a number of studies have failed to find a beneficial effect[51,52]. Recently, it has emerged that when we eat is as important as what we eat. Large epidemiological studies have shown that reduced prolonged nightly fasting and/or reduced nighttime eating is associated with improved glycemic regulation, a reduction in the risk of breast, prostate cancer, and pancreatic cancer, and of breast cancer recurrence[28,35,36,53,54]. Interest in TRE as a non-pharmacological strategy to improve metabolic health in obese humans has thus increased in recent

**Fig. 8 TRF normalizes gene circadian rhythms in liver, mammary fat, and breast tumors. a** QPCR analysis of the core clock genes in liver, MFP, and tumor samples from mice on normal chow collected at different times (ZT0, 4, 8, 12, 16, 20) over a 24 h period. ZT0 indicates 6 am and the start of the light phase. Data is plotted over 48 h to facilitate visualization of the circadian rhythms. Gray rectangles indicate period of lights off. **b** QPCR analysis of the clock genes *Bmal1, Per1,* and *Cry1* in liver samples collected at different times over 24 h in mice on ad libitum HFD (AL—orange), time-restricted HFD (TRF—green) or chow diet (NC—blue). Data are presented as mean normalized expression ± SEM (*n* = 4 mice/time/group). **c** QPCR analysis of the clock genes *Bmal1, Per1,* and *Cry1* in MFP samples over 24 h in mice on ad libitum HFD or TRF or normal chow diet (*n* = 4 mice/time/group), colors as in **b. d** QPCR analysis of the clock genes *Bmal1, Per1,* and *Cry1* in tumor samples collected over 24 h (*n* = 4 mice/time/group), colors as in **b**. Data in **b, c,** and **d** are plotted over 48 h to facilitate visualization of the circadian rhythms. **e** QPCR analysis of the clock genes *Bmal1, Per1,* and *Cry1* in Py230 cells following a 50% serum shock without (HS–blue) or with 10 nM insulin (HS + ins–red) collected at different times over 48 h (*n* = 3/time/group). Dotted line indicates a common shared curve that explains the variation in the data. Data are presented as mean normalized expression (*n* = 4/time/group), error bars are omitted for clarity. Twenty-four hour rhythms were analyzed statistically using RAIN with a period of 24 ± 4 h, and circadian curves fitted using PRISM using the equation Y = baseline + amplitude*cos(frequency*X + phaseshift) using a frequency of 0.2618 for a 24 h period. Genes with statistically significant circadian rhythms are shown as curves, acyclic genes are shown as horizontal lines. Statistical data and circadian parameters are given in Supplementary Table 1.

years due to the relative ease of compliance for the maintenance of long-term benefits[37]. A small study of TRE for 12 weeks in 19 individuals with metabolic syndrome demonstrated improved cardiometabolic health and, importantly, reported good long-term compliance with the TRE intervention months after the end of the trial[42]. Furthermore, although TRE is not designed to have caloric restriction, many studies report weight loss, loss of visceral fat mass, as well as improved metabolic measures in obese individuals[43]. Thus, although clinical studies of TRE in humans have been very small in size and under-powered for most outcomes, they are suggestive of positive metabolic improvement[29,55] and TRE is a potential lifestyle intervention, which can improve metabolic health. It is noteworthy that meal timing has also been shown to improve circadian rhythms in cardiovascular diseases, Hutchinson's disease (HD), and pancreatic tumors[23,54,56,57]. In the mouse model of HD, the circadian alignment of meal timing (feeding access 9 pm to 3 am) resulted in improvements in locomotor activity rhythm, sleep awakening time, autonomic nervous system dysfunction, and motor performance[56]. In addition, in the mouse model of pancreatic adenocarcinoma, meal timing from 8 am to 12 pm resulted in reduced tumor growth compared with ad libitum feeding of fat diet[54]. Reinforcement of the host circadian timing system with meal timing induced 24-hour rhythmic expression of clock-controlled genes, which translated into cancer growth inhibition. Interestingly, a pilot study in 11 obese humans for 4 days demonstrated that eating between 8 am and 2 pm (early TRE) showed greater improvement of circadian rhythms compared with the control schedule (8 am to 8 pm)[58] and a similar study of 15 prediabetic men with continuous glucose monitoring showed that early TRE reduced mean fasting glucose[45]. Eating at night is particularly detrimental as it is associated with higher BMI and increases the risk of breast and prostate cancer[36,53,59]. These findings underscore the importance of meal timing that is synchronized to circadian rhythms to maximize improvement in overall metabolism. Despite these findings, there have been very few interventional studies in animals and, to our knowledge, none in humans to study the effect of TRF on obesity-driven tumorigenesis[60,61]. Furthermore, we are unaware of any study that has tested the effect of TRF on obese postmenopausal breast cancer.

In the present study, we showed that TRF improved metabolic health in obese postmenopausal mice on a HFD without altering nutrient intake, and resulted in a striking inhibition of tumor initiation, tumor progression, and metastasis compared with an ad libitum HFD. Meal timing or TRF in lean mice has also been shown to attenuate Glasgow osteosarcoma and P03 pancreatic cancer in B6D2F1 mice[34,54], PyMT tumor growth in FVB mice[60], and lung metastasis in the Lewis lung carcinoma mouse model[61] but these effects were unrelated to alterations in metabolism

associated with obesity and the mechanism by which TRF was acting in these descriptive studies was not determined. Numerous studies indicate that obesity drives metabolic dysregulation in both male and female mice[18,40,47,62] and increases the risk for various cancers in both men and women[7,63]. Notably, early eating reduces the risk of both prostate and breast cancer[36]. Although the present study focused on obese, female, postmenopausal mice, we anticipate that similar beneficial effects of TRF will be observed in obesity-driven cancers in males.

If TRE is to be proposed as a lifestyle intervention, it is important to understand the molecular mechanisms underlying the reported beneficial effect. Obesity induces insulin resistance and compensatory hyperinsulinemia, and hyperinsulinemia is linked to an increased risk of cancer[64,65], particularly in postmenopausal breast cancer patients with an aggressive, metastatic phenotype, and poor prognosis[14,17,66]. Insulin is a potent mitogen for mammary epithelial cells, increasing terminal end bud formation and branching in virgin mammary glands, and causing ductal hyperplasia[13,67]. A direct mitogenic role is supported by studies from other groups, as well as our in vitro studies, which report that insulin promotes proliferation, migration, and invasion of breast cancer cells and promotes breast cancer progression[13,67–70]. Our studies using exogenous insulin to induce hyperinsulinemia in vivo demonstrate that insulin promotes breast cancer growth in the context of obesity. A similar effect of hyperinsulinemia on breast tumor growth and progression in the female MKR mouse model for hyperinsulinemia has been reported[71,72]. Furthermore, reducing hyperinsulinemia using DZ reduced tumor growth in obese mice suggesting a role for hyperinsulinemia in tumor development and progression. Studies using a β-3 adrenergic receptor agonist to reduce circulating insulin levels or using a tyrosine kinase inhibitor that inhibited both the IR and IGF-1R reported reduced the tumor growth in MKR mice, further supporting our in vivo observation with DZ[71–73]. Our study demonstrated that TRF effectively reduced insulin levels, improved insulin sensitivity and slowed tumor growth. Importantly, the reduction of tumor growth in the TRF group being abrogated by artificial insulin augmentation by insulin pump, suggests that TRF corrected obesity-induced hyperinsulinemia, and that this insulin-lowering effect was the mechanism by which tumor growth was inhibited in these postmenopausal breast cancer models.

A number of insulin-lowering drugs have been shown to have tumor inhibitory effects in various mouse models of cancer[71–74]. For instance, reversal of hyperinsulinemia by a liver-specific mitochondrial protonophore is sufficient to reverse the obesity-induced acceleration of tumor growth[74]. Clinical trials of many drug candidates have been somewhat disappointing however[65], so a consideration of alternative strategies for reducing hyperinsulinemia in

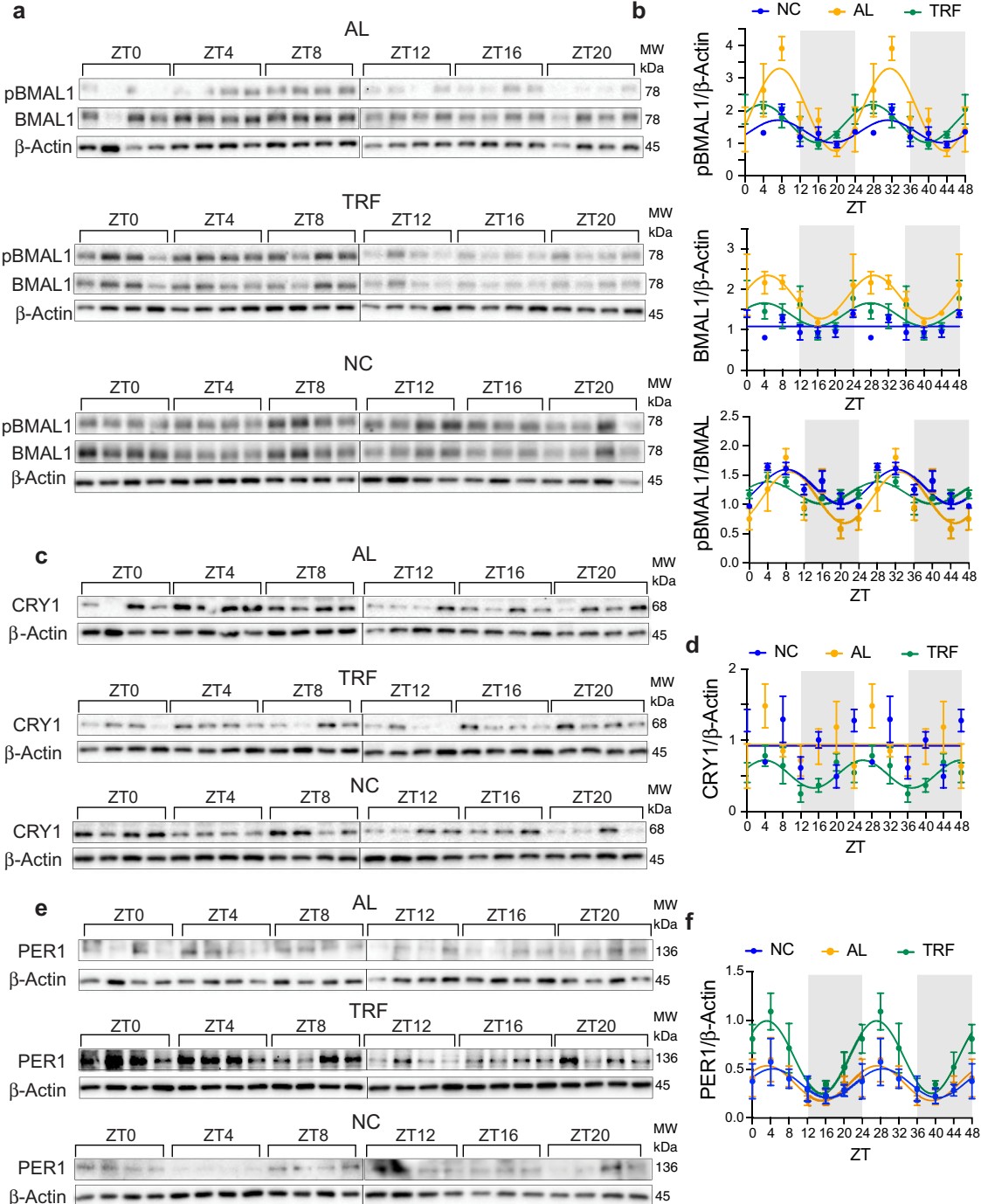

**Fig. 9 TRF normalizes protein endogenous circadian rhythms in breast tumors. a** Western blot analysis of the clock protein BMAL1 and phospho-BMAL1(Ser42) in tumor samples collected at different time points (ZT0, 4, 8, 12, 16, 20) over a 24 h period in mice on ad libitum HFD (AL—orange), time-restricted HFD (TRF—green), or chow diet (NC—blue). ZT0 indicates 6 am and the start of the light phase. **b** Quantification of the normalized phospho-BMAL1(Ser42), total BMAL1, and the phospho-BMAL1/BMAL1 ratio is graphed over 48 h to facilitate visualization of rhythms ($n = 4$ mice/time/group). Gray rectangles indicate period of lights off. **c** Western blot analysis of the protein CRY1 in tumor samples collected at different time points (ZT0, 4, 8, 12, 16, 20) over a 24 h period in mice on ad libitum HFD, TRF or NC. **d** Quantification of the normalized CRY1 protein expression ($n = 4$ mice/time/group). **e** Western blot analysis of the clock protein PER1 in tumor samples collected at different time points over a 24 h period in mice on ad libitum HFD, TRF, or NC. **f** Quantification of the normalized PER1 protein expression ($n = 4$ mice/time/group). Data are presented as mean ± SEM normalized to β-actin. Twenty-four hour rhythms were analyzed statistically using RAIN with a period of 24 ±4 h, and circadian curves fitted using PRISM using the equation $Y = \text{baseline} + \text{amplitude}*\cos(\text{frequency}*X + \text{phaseshift})$ using a frequency of 0.2618 for a 24 h period. Proteins with statistically significant circadian rhythms are shown as curves, acyclic proteins are shown as horizontal lines. Statistical data and circadian parameters are given in Supplementary Table 2.

cancer therapy is urgently needed. In the context of our findings and the pilot studies in humans demonstrating that TRE is an effective and simple dietary intervention to improve insulin sensitivity[44,58], we suggest TRE as potential strategy for reducing hyperinsulinemia and thus tumor growth in cancer patients.

Obesity also disrupts circadian rhythms in peripheral tissues, and numerous studies have reported a strong association between circadian rhythm dysregulation and cancer progression[25,75,76]. We observed a similar disruption in the liver and MFP circadian clocks in obese mice, as mice with ad libitum access to HFD eat continually over 24 h and did not show normal circadian feeding behavior[40]. TRF effectively enforces more natural feeding behavior and corrected HFD-associated circadian rhythm disturbances[40,47]. Tumors have been reported to have disrupted circadian rhythms[15] and indeed we observed unusual rhythms with in-phase cycling of the *Clock, Per1, Per2, Cry1,* and *Reverba* genes in mammary tumors. Many of the tumor circadian rhythms of genes/protein expression were suppressed by ad libitum diet-induced obesity compared with lean controls but were restored by TRF to patterns similar to those in the normal tissues. These findings indicate that the tumor clocks are sensitive to nutrient signals and to food intake timing. We demonstrated that the Py230 cells exhibit circadian rhythms in the core clock genes in vitro following a serum shock, which is consistent with rhythms observed in the tumors. Interestingly, the phases of the oscillations in the tumors were different from the non-tumor tissues such as the liver and MFP. As expected, *Reverba, Per,* and *Cry* genes were out of phase with *Bmal1* in the non-tumor tissues[23] with these negative clock regulators being induced 10–14 h after *Bmal1*, which is required for the generation of the 24 h circadian rhythm. In tumors, the induction of *Reverba*, and the *Per* and *Cry* genes occurred much faster than in the non-tumor tissues, within 6 h of the peak *Bmal1* expression. There was also induction of *Clock* oscillations that were in-phase with the negative regulators. Interestingly, phosphorylation of BMAL1 at Ser42 and total BMAL1 protein showed a circadian rhythm, so it is possible that the clock in the tumor is driven by post-translational modifications. Our data show a role for insulin in driving tumor growth in obesity, and insulin has also been shown to cause a phase advance in clock genes[77] and to act as a circadian synchronizer[78–80]. Insulin also induces Akt-dependent Ser42 phosphorylation of BMAL1, which promotes nuclear exclusion and repression of BMAL1 transcriptional activity[81]. The lack of cyclicity of PER1 and CRY1 in the tumors from AL HFD mice is consistent with repression of BMAL1 activity, which is restored by TRF. However, further studies will be necessary to test this model.

In summary, we confirmed and extended prior reports that TRF normalizes hyperinsulinemia, improves metabolic deregulations, and restores circadian rhythms. We found that TRF inhibited tumor growth, delay tumor initiation, and reduce breast cancer metastasis to lung in models of postmenopausal obesity-driven breast cancer. Here, we provide a molecular rationale for the adoption of TRF to reduce insulin resistance and normalize circadian rhythms against obesity-driven breast cancer. These findings support the possibility that a simple dietary manipulation through TRE might be a practical way to achieve a beneficial effect in breast cancer therapy. On this basis, clinical trials to determine the relevance of TRE to human breast cancer are warranted. Thus, TRE could be a potential non-pharmacological intervention to prevent or inhibit the progression of human breast cancer.

## Methods

**Cell culture**. Py230 (RRID: CVCL_AQ08; female) mammary tumor cell line was isolated by LGE[49]. The cells were maintained in F12K medium containing 5%

FetalClone II, 0.1% MITO + serum extender, gentamycin (50 μg/ml), and amphotericin b (2.5 μg/ml) at 37 °C in a humidified 5% $CO_2$ incubator. E0771 (RRID:CVCL_GR23; female) mammary tumor cell line (SKU: 940001, CH3BioSystem, Amherst, NY)[82] was maintained in RPMI 1640 medium supplemented with 10 mM HEPES, 5% FBS, penicillin (100 units/ml) and streptomycin (100 μg/ml) at 37 °C in a humidified 5% $CO_2$ incubator. Chemicals and reagents were purchased from Sigma-Aldrich (St. Louis, MO)

**In vivo mouse studies**. All mice were housed at 24 ± 2 °C on a 12 h light/12 h dark cycle (light on at 6 am) and the animals had free access to food and water prior to being placed in study groups. Mice were placed on an HFD (D12492, 60% kcal from fat; 5.24 kcal/g, Research Diets Inc, New Brunswick, NJ) to cause obesity. Control mice were fed NC (rodent 5001). Group sizes were chosen according to pilot experiments and female mice randomly assigned to groups. Adult C57Bl/6 J female mice (Jackson Labs, Sacramento, CA) were used for all studies.

**Postmenopausal mouse models**. Menopause was induced in female mice either surgically or chemically. Female mice aged 7–8 weeks were surgically ovariectomized (OVX) under isoflurane anesthesia. For the chemically-induced ovotoxicity model, mice aged 4 weeks were injected daily with VCD (160 mg kg⁻¹ day⁻¹ in vegetable oil, i.p.) for 15 days and maintained without further treatment for another 95 days prior to experimental use. VCD causes the loss of ovarian small follicles (primary and primordial) in mice and rats by accelerating the natural process of atresia and can be used to mimic human menopause in rodent models of disease[83,84]. Female PyMT transgenic mice on the C57Bl/6 J background[85] were used as a spontaneous tumor model and OVX at 7 weeks of age.

**Orthotopic cancer cell injection**. Py230 cells ($1 \times 10^5$) and E0771 cells ($12 \times 10^3$) were injected in 20 μl matrigel bilaterally into the #3 and #4 MFP after the 3rd or 6th week of TRF intervention, respectively (Fig. S4a and Fig. S5a). For the pre-existing tumor inhibition study, Py230 cells ($1 \times 10^5$) were injected into the MFP of mice on 10 weeks of HFD feeding. Mammary glands were assessed 3 weeks later by ultrasound to verify the presence of tumors. TRF was then initiated following 3 weeks of tumor cell injection and tumor volumes were measured weekly with calipers. In lung metastasis experiments, E0771 cells ($5 \times 10^5$) were injected into the tail vein after the 4th week of TRF intervention.

**Time-restricted feeding**. A schematic representation of the TRF strategy is outlined in Figs. S1a, S4a, 4a, 5a, and S5a. In brief, 20 OVX C57BL/6 mice were started on the HFD at 10 weeks of age. Body weights and food intake were measured weekly thereafter. When the HFD mice reached an average weight of 45 g (~10 weeks of HFD), the HFD group was divided into AL and TRF groups of 10 mice each. Custom fully automated feeder cages (Fig. S1a) were used for the AL and TRF intervention study in which the AL group had access to the diet at all times, whereas the TRF group had access to the HFD from 10 pm to 6 am daily (Fig. S1a). A group of OVX C57BL/6 mice was maintained on the NC diet as a lean control group. For studying the effect of TRF on tumor growth in lean mice, 20 mice at 20 weeks of age were grouped into NC-AL and NC-TRF groups of 10 mice each. In the VCD mouse model, at day 110 from initiation of VCD treatment, mice were started on the HFD. When they reached an average bodyweight of ~45 g, the mice were subjected to the TRF intervention as described above. All tissue was collected between 10 am and 12 pm, i.e. zeitgeber time (ZT) 4–6. For the tumor inhibition study, OVX C57BL/6 mice were started on an HFD at 9 weeks of age. Following 10 weeks on HFD Py230 cells were injected, then 3 weeks later the mice were divided into HFD ad libitum and TRF groups or NC group (12–14 mice/group) (Fig. 5a). For the circadian analysis, tumor, MFP, and liver samples were collected every 4 h over a 24 h period for the measurement of clock gene and protein expression by qPCR and western blotting, respectively.

In the transgenic mouse model, 20 PyMT female OVX mice at 7 weeks of age were OVX and randomly divided into HFD-AL or HFD-TRF groups (Fig. 4a). Bodyweight was measured weekly and food intake noted daily to assess the caloric intake per mouse. Mammary glands were assessed weekly by ultrasound and when tumors became palpable tumor volumes were measured weekly with calipers. Tumor volumes were calculated using the following equation: $0.52 \times$ long diameter × short diameter². At euthanasia, plasma was collected and tumors, MFPs, visceral (perigonadal) fat pads, liver, spleen, and lungs were harvested and weighed. Tissues were formalin-fixed or frozen for further analysis. All tissues were collected at ZT4–6.

**Ultrasonography methods**. A VevoMD ultra high-frequency ultrasound system equipped with a 70 MHz probe (FUJIFILM VisualSonics, Toronto, Canada) was used to image mammary glands. Animals were sedated in a supine position under isoflurane anesthesia delivered via a nose cone. The ventral fur was removed with chemical depilatory cream, each of the nipples was labeled with an indelible marker, and ultrasound gel was applied. Nipple location was confirmed by ultrasonography. Nipples appeared as superficial submillimeter foci with a hyperechoic deep margin. If present, breast masses adjacent to the nipples were identified. They appeared as subdermal, well-circumscribed, hypoechoic, and occasionally cystic masses. They are highly vascular on Doppler ultrasound. Masses were measured in

two axes in the transverse plane and recorded. For the tumor inhibition study, orthotopic tumor-bearing mice were imaged at 3 weeks post Py230 tumor cells injection. The transgenic PyMT mice were serially imaged biweekly starting at 9 weeks of age to 15 weeks of age.

**Insulin pump experiments.** For the artificial hyperinsulinemia experiments, we injected Py230 cells into the thoracic and inguinal MFPs bilaterally in OVX mice on ad libitum NC then 3 weeks later insulin pumps (AP 2006, DURECT Corporation, Cupertino, CA) were implanted subcutaneously in the interscapular region to deliver a constant low level of insulin (0.2 U/day). This level of insulin did not cause hypoglycemia. Control mice were implanted with pumps containing normal saline solution. For the more aggressive E0771 cells, we implanted insulin pumps first and 2 weeks later injected the cells. To test whether elevating the insulin in obese mice would cause greater tumor growth, obese mice on ad libitum HFD were injected with Py230 cells then implanted subcutaneously with insulin pumps delivering a constant dose of 0.6 U/day to further enhance hyperinsulinemia, or control pumps containing saline, to further enhance hyperinsulinemia (Fig. 7h). We also implanted insulin pumps delivering 0.3 U/day in mice on HFD-TRF before they were injected with Py230 cells (Fig. 7o). After 4 weeks, pumps were replaced with new insulin pumps delivering 0.3 U/day. Tumor volume and fasting glucose were measured over time. At the termination of the experiment, tumor weights and volumes were measured. An NC group as mentioned previously was used for comparison.

**DZ experiment.** To reduce insulin secretion, we treated obese mice on ad libitum HFD with DZ, a potassium-ATP channel activator used clinically to reduce hyperinsulinemia. The mice were injected with Py230 as in previous experiments, then three weeks later we switched a group of 10 mice to HFD containing DZ (1.125 g/kg,) whereas the remaining 10 mice were maintained on HFD as controls (Fig. 7i). Tumor volume and fasting glucose were measured over time. At the termination of the experiment, tumor weights, and tumor volumes were measured.

**Glucoregulatory assessments.** For the glucose tolerance test (GTT), 8–10 mice per group were fasted for 6 h prior to GTT. Blood glucose was measured by tail bleed at 0 min and then 1 g/kg glucose injected intraperitoneally. Blood glucose was monitored at intervals up to 120 min using a glucose meter (Easy Step Blood Glucose Monitoring System, Home Aide Diagnostics, Inc., Deerfield Beach, FL). For the insulin tolerance test (ITT) mice were fasted for 6 h and blood glucose measured at 0 min prior to injection of 0.65 U/kg insulin intraperitoneally. Blood glucose was measured at intervals up to 120 min. For the pyruvate tolerance test mice were fasted for 6 h and blood glucose measured at 0 min prior to injection of 1 g/kg pyruvate intraperitoneally. Blood glucose was measured at intervals up to 120 min. Terminal fasting glucose was also measured following 6 h fasting. Terminal fasting plasma insulin was measured using the mouse/rat insulin kit (Meso Scale Diagnostics, Rockville, MD) and the HOMA-IR was calculated as follows: (fasting serum insulin concentration (mU/ml)) × (fasting blood glucose levels (mg/dl))/(405).

**Blood ketone levels and glucose–ketone index.** Blood glucose and ketone levels were measured at 3 h intervals from 6 AM to 12 AM by tail bleed. Blood ketones were measured using a ketone meter (Abbott Precision Xtra® Blood Glucose Meter & Ketone Monitoring System, Alameda, CA). The glucose–ketone index was calculated by dividing the blood glucose level (mM) by the ketone level (mM).

**Hepatic steatosis.** Hepatocellular steatosis was assessed on 5 μm hematoxylin and eosin-stained (H&E) formalin-fixed, paraffin-embedded tissue sections by scanning cross-sectional areas on five independent slides by bright-field light microscopy (Eclipse TE300, Nikon, New York, NY) at ×20 magnification. The number and area of lipid droplets (stained negative for H&E) were measured using Image J software[86].

**Immunoblotting.** Mouse tissues were dissected, homogenized and sonicated in radioimmunoprecipitation assay lysis buffer (150 mM NaCl, 50 mM Tris·HCl pH 7.4, 1% NP-40, 0.25% sodium deoxycholate, 1 mM EDTA, 1 mM sodium orthovanadate, 10 mM NaF, 10 mM glycerophosphate, 5 mM sodium pyrophosphate) containing protease and phosphatase inhibitors. Lysates were incubated for 30 min on ice and centrifuged for 20 min at 10,000 rpm at 4 °C. The protein content of tissue lysates was quantified using the micro BCA protein assay kit (23225, ThermoFisher Scientific, New York, NY) using BSA as standard. Total protein lysates (20 μg) were separated on 10–15% polyacrylamide gels and transferred onto Polyvinylidene difluoride membranes (IPVH00010, Millipore Sigma). Membranes were blocked in 5% non-fat milk or BSA in PBS with 0.05% tween 20 (PBST) for 2 h at RT and then incubated at 4 °C overnight with the primary antibody. Next, the membranes were washed three times with PBST and incubated with HRP-conjugated secondary antibodies (Santa Cruz Biotechnology, Santa Cruz, CA) for 40 min. Signals were captured on X-ray films or ChemiDoc Imaging System (BioRad, Hercules, CA) following the reaction with the enhanced chemiluminescence substrate (SuperSignal™ West Pico PLUS Chemiluminescent Substrate,

ThermoFisher Scientific). β-Actin was used as loading control. For western blotting: anti-phospho-AKT (Thr308, 4056), anti-phospho-AKT (Ser473, 9271), anti-panAKT (9272), anti-phospho-GSK3β (Ser9, 9336), anti-GSK3β (9315), anti-β actin (3700) antibodies, anti-phospho-ERK1/2 (Thr202/Tyr204, 4377), anti-ERK1/2 (4695) and phospho-BMAL1 (Ser42, 13936) were purchased from Cell Signaling Technology (Danvers, MA). Antibody to PER1 (ab136451), CRY1 (ab104736), and BMAL1 (ab 93806) were purchased from Abcam (Cambridge, MA). Antibodies were used according to manufacturers' recommendations. Uncropped source blots are included in the Supplementary Materials.

**RNA isolation and real-time PCR.** RNA was isolated from frozen liver, MFP, and tumor tissue using the RNeasy Mini Kit (Qiagen, Germantown, MD) and reversed transcribed into cDNA using the qScript cDNA synthesis kit (Quantabio, Beverly, MA). Real-time PCR was performed on a Bio-Rad CFX384™ real-time PCR detection system using iTaq™ Universal SYBR® Green Super mix (BioRad). The primers are listed in Supplementary Table 3.

**Immunohistochemistry.** Slides of tumor sections were de-paraffinized with xylene and rehydrated with ethanol along a concentration gradient of 100 − 70% followed by washing with distilled water. The slides were then heated to 105 °C in antigen retrieval solution (10 mM Sodium citrate, 0.05% Tween 20, pH 6.0) for 20 min and were allowed to cool at room temperature followed by washing with distilled water. Sections were then incubated with 0.3% hydrogen peroxide for 10 min at room temperature to remove the endogenous peroxidase activity and then blocked with 2% BSA in PBS for 1 h at 37 °C. After blocking for 1 h, sections were incubated with a rabbit polyclonal anti-Ki67 antibody (15580, Abcam, 1:2000 dilution, 2 h at RT), rabbit polyclonal anti-CD31 antibody (28364, Abcam, 1:1000 dilution, overnight at 4 °C). After rinsing three times in PBST, the sections were incubated with biotinylated secondary universal horse anti-rabbit IgG for 30 min, followed by incubation with avidin−biotin peroxidase complex for 30 min at room temperature. Sections were washed three times with PBST, exposed to diaminobenzidine, washed, and counterstained with Meyer's hematoxylin. Photographs of the tumor sections were captured using a Nanozoomer 2.0HT Slide Scanner (Hamamatsu Photonics, Bridgewater, NJ).

**Lung whole-mount staining.** Mice were perfused through the right ventricle with PBS in order to remove blood from the lungs. Lungs were subsequently inflated with Methyl Carnoy's fixative and fixed for two days at RT in fixative. Next, the tissue was rehydrated and stained with Carmine staining solution for 2 h and de-stained in acid ethanol. The lungs were then dehydrated, transferred to methyl-salicylate solution, and photographed.

**Cell viability measurement.** Py230 cells were treated with different concentrations of insulin in 2% serum containing medium for 24 h and cell viability was assessed by Trypan blue staining assay following the manufacturer's instructions. For testing the effect of steroid hormones on breast cancer cell proliferation, Py230 cells were plated at 50,000 cells per well into 24-well plates. Triplicate cultures were treated with ß-estradiol (0, 1, 10, or 100 pM) or progesterone (0.1, 1, 10 nM) with or without 10 nM insulin (hatched bars) for 48 h. Cells were washed, trypsinized, and counted using a coulter counter. Data were analyzed using one-way analysis of variance (ANOVA) or unpaired *t* tests to evaluate the addition of insulin to the cultures. For testing the effect of ketone bodies (β-hydroxybutyrate) on breast cancer cell proliferation, Py230 and E0771 cells were treated with different doses of β-hydroxybutyrate in 5% or serum-free medium containing different glucose concentration. Cell viability was assessed as previously mentioned.

**Migration assay.** To investigate the effect of insulin treatment on the migratory properties of Py230 cells, a scratch assay was performed. In brief, $5 \times 10^5$ cells were seeded in six-well plates and were allowed to attach and reach 80% confluency. Cells were then incubated with starvation medium containing 2% fetal calf serum (FCS) for 6 h followed by treatment with different doses of insulin or only starvation medium (which served as control) for 2 h. After the incubation period a scratch was made through the cell monolayer using a 200 μl pipette tip. Cells were then washed with PBS and images of the scratch area were taken using an inverted phase contrast microscope with ×20 objective. For each well, three different scratched areas were photographed and their location on the plate was marked. The cells were further incubated for 16 h in starvation medium and the same scratched areas were re-photographed. The gap distance of the wound was measured in the scratched area at both time points. Data were quantified by measuring the total distance from the edge of the scratch towards the center that was traveled by cells before and after treatment

**Invasion assay.** Invasion is another characteristic property of metastasis and the effects of insulin treatment on the motility of Py230 and E0771 cells were evaluated using a Matrigel-coated transwell chamber with 8 μm pores (Sigma-Aldrich). In brief, $2 \times 10^5$ cells were seeded in six-well plates and were allowed to attach overnight. Cells were then incubated with starvation medium containing 2% FCS for 6 h followed by treatment with different doses of insulin or only starvation

medium (which served as control) for 2 h. Following the incubation period, cells were collected and reseeded at a density of $0.5 \times 10^5$ live cells into the upper well of the chamber containing serum-free culture medium. The lower well of the chamber was filled with complete culture medium and incubated for 24 h. Thereafter, cells present on the upper side of the transwell chamber were gently removed and cells in the lower part of the chamber were fixed with 4% paraformaldehyde and stained with 0.5% crystal violet. The images of transmigrated cells were then captured using an inverted phase-contrast microscope (Eclipse TE300, Nikon, Melville, NY) with ×20 objective and counted.

**Spheroid formation assay**. In brief, $2 \times 10^5$ Py230 cells were seeded in ultra-low adherence SIX-well plates and treated with different doses of insulin or only starvation medium alone (which served as control). The media was changed every 3 days and after 7 days the spheroids formed were imaged using bright-field microscope with ×10 magnification and counted.

**Clock gene expression with insulin treatment**. In brief, $3 \times 10^5$ of Py230 cells were seeded in six-well plates and were allowed to attach and were incubates with serum-free medium for overnight. Cells were then incubated with 2% FCS medium or medium with 10 nM of insulin for different time points over 8 h. In next set of experiment, $3 \times 10^5$ of Py230 cells were seeded in six-well plates and were allowed to attach. Cells were then incubated with serum-free medium for overnight. Next day the cells were serum-shocked by replacing medium with 50% horse serum for 2 h, then washing twice with PBS. The cells were then incubated with medium containing 2% charcoal-treated serum with or without 10 ng/ml of insulin for different time points over 48 h. Following incubation, RNA was isolated and mRNA expression of different clock genes was analyzed by real-time PCR.

**Statistical analysis**. Data are presented as mean ± standard error of the mean. Data were compared using a two-way ANOVA for independent measures, with group factor and time factors in cases of weekly data, followed by Bonferroni post hoc tests with a significance threshold set at $P < 0.05$. The one-way ANOVA was done with the significance threshold set at $P < 0.05$. Data not meeting the criteria for normality were analyzed using the non-parametric Mann–Whitney test. Statistical analyses were performed with PRISM8 (GraphPad, La Jolla, CA) and graphs were made with SigmaPlot 10.0 (Systat Software Inc, San Jose, CA). Circadian gene expression over 24 h was analyzed using the software RAIN[87] and genes showing circadian rhythms were visualized by curve fitting in PRISM8. The data were fit using the equation $Y = \text{Baseline} + \text{Amplitude}*\cos(\text{Frequency}*X + \text{Phaseshift})$ using a frequency of 0.2618 for a 24 h period. For each curve, the fit was compared with a constant expression level (non-circadian). We also tested whether a single curve could account for all the expressions in all three groups (NC, AL, and TRF). The expression levels and associated curves were graphed over 48 h to facilitate visualization of the rhythms.

**Reporting summary**. Further information on research design is available in the Nature Research Reporting Summary linked to this article.

## Data availability

All the data supporting the findings of this study are available within the article and its supplementary information files and from the corresponding author upon reasonable request. A reporting summary for this article is available as a supplementary information file. Source data are provided with this paper.

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

## Acknowledgements

We thank histology and microscopy cores at the Moores UCSD Cancer Center for their assistance. Funding: this work was supported by National Institutes of Health grants CA196853, CA155435, CA023100, VA Merit Review grants I01BX002709 and I01BX004848, and a VA Senior Research Career Scientist IK6BX005224 award. The data that support the findings of this study are available from the corresponding author upon reasonable request. Copyright-free cartoon images were obtained from publicdomainvectors.org.

## Author contributions

M.D. designed the studies, performed the animal studies, the tissue staining, and qPCR studies, wrote the first draft of the manuscript and prepared drafts of the figures. L.G.E. assisted with the design and performance of animal studies, tissue staining, and assisted with drafting and editing the manuscript. D.K., A.O., and C.S. assisted with animal studies and tissue processing. A.O. and M.K. performed animal husbandry and diet administration. E.G. performed the tail vein injections of cells and analyzed the data for metastasis of breast tumor to lung. I.G.N. and T.M. performed the ultrasonography of spontaneous breast tumors in mice. N.J.G.W. developed the hypotheses, designed the experiments, provided

financial support, supervised all experiments, edited and proofed the manuscript, and generated final figures. All authors have approved the final manuscript.

## Study approval

All animal experiments were approved by the UC San Diego Institutional Animal Care and Use Committee (IACUC) and were conducted according to the NIH guidelines for the Care and Use of Laboratory Animals.

## Competing interests

The authors declare no competing interests.
