## [Peer Review File · Nature Communications]

Reviewers' comments:

Reviewer #1 (Remarks to the Author):

The manuscript "Time-restricted feeding normalizes hyperinsulinemia to inhibit breast cancer in obese postmenopausal mouse models" by Das et al. includes the data presented at ENDO 2019 meeting (J Endocr Soc. 2019 Apr 15; 3 Suppl 1) published online. It is well established that calorie restriction (CR) reduces the progression of tumors in various animal models, although the efficacy of CR may be limited to certain cancers. This study narrows in on post-menopausal breast cancer initiation and progression in the context of time-restricted feeding (TRF) , a time (but not caloric)-restricted approach to energy intake which improves metabolism in a number of tissues and systemically under conditions of high fat diet feeding. The manuscript is well written and addresses a clinically important issue. By using HFD-fed postmenopausal mouse models, the authors show that TRF reduces obesity and improves metabolic dysregulation in obese reduces Py230 mammary tumor growth, attenuates E0771 mammary tumor growth and metastasis to lung, and inhibits tumor initiation and growth in obese OVX-PyMT transgenic mice concomitantly with normalizing endogenous circadian rhythms in breast tumors. They provide evidence suggesting that TRF acts through modulating hyperinsulinemia. Implication of insulin is expected, as obesity-related insulin resistance is a well known risk factor for breast cancer development in post-menopausal women. However, injected insulin was not found to affect cancer risk in clinical trials, and it is clear how this study adds to resolving this controversy.

The overall effects of TRF on tumor growth and metastasis are convincing. However, there is a lack of sufficient detail related to the mechanisms involved. The lack of detailed attention to zeitgeber times, as well as the absence of comparison to normal breast tissue, limit interpretation in a number of areas. The links between insulin signaling and clock proteins (BMAL1 specifically) should be addressed both experimentally and in the text. Specific comments:

1. The authors should address whether OVX or VCD treatment affects overall rhythmicity of the animal, considering the focus of the study is on time-restricted feeding. How diurnal behavior is affected with and without TRF is not discussed, nor how it compares between the two different models.
2. Related to point #1, can the authors provide the total food intake presented in Figure 1 into day vs. night energy intake? Though ad libitum-applied HFD has been proposed to reduce rhythmicity of energy intake, this is somewhat controversial. It would be nice to see the extent to which the nice tumor reduction effects have anything to do with restored rhythms in energy intake.
3. The authors do not discuss in sufficient depth the relationship between insulin signaling and the circadian clock. TRF has been reported to function in part by restoring rhythmicity in various organs. BMAL1 itself is downstream of IR, getting phosphorylated directly by AKT and GSK3B. The results of Figure 2K suggest that BMAL1 stability would be altered under TRF vs. AL and NC. Is this the case? I

think that additional analysis of circadian molecules in the breast stroma and tumor tissue would indicate whether TRF functions within the tumor to affect core clock proteins vs. is functioning in a largely clock-independent manner. Rather than simply showing gene expression in Figure 7, BMAL1 protein should be directly analyzed, particularly its phosphorylation as antibodies are available and widely used for such analysis. This is a critical point, as BMAL1 has emerged as a key molecule in several tumor types, with under- or over-expression having different effects depending on the tumor type.

4. The authors discuss Cidea and Cidec as being elevated in ad libitum-fed mice compared to TRF. These are highly dynamic molecules in the context of high fat diet feeding in several tissues. The authors do not state at which zeitgeber time these are being analyzed.

5. Figure 7 addresses the rhythmicity of clock genes in tumors of normal chow, ad-libitum HFD-fed mice, and TRF HFD-fed mice. What is very confusing about these data is that Per and Bmal1 are in the same phase. (In fact, all of the clock genes shown are in the same phase, suggesting the possibility of some normalization artifact.) Though Per1 and Bmal1 are generally anti-phasic in expression, the authors state that TRF normalized rhythms to normal in the tumor tissues. These are in no way “normal”- at least not compared to healthy cells. Here, it would be helpful to have the non-tumor tissue as a comparison. As presented, these data are very difficult to interpret what is going on with the clock in the various conditions. To the authors’ credit, this issue is finally discussed briefly in the discussion section. I think some experimental approach (i.e. comparing tumor to normal breast stroma) would be a better way to address this strange result. In addition, amplitude of oscillation could be compared between tumor and non-tumor.

Finally, it has been reported that diets rich in saturated and omega 6 fatty acids are pro-inflammatory and increase breast cancer risk, while diets rich in omega 3 fatty acids are anti-inflammatory and decrease it. It would be interesting to test if/how the link of dietary intake with cancer initiation and progression is affected by diet composition, and specifically by the types of fatty acids in HFD.

Reviewer #2 (Remarks to the Author):

Summary to Author:

This timely, novel, and well-designed study provides intricate detail into a complex area of research. Strengths include characterizing the metabolic effects of TRF in both tumor-free and tumor-bearing surgical and chemically induced ovariectomized mice, and the use of multiple mammary tumor models (orthotopic and transgenic), numerous techniques and “break and fix” methods, and time course outcomes to identify circadian rhythm genes altered with HFD and responsive to TRF within the tumor microenvironment. In total, the data presented in this manuscript paints a very thorough

picture and provides new insights into how a TRF paradigm can normalize high-fat diet-induced insulin levels to reduce tumor outgrowth in preclinical models of postmenopausal breast cancer. Interestingly, the TRF benefits on normalizing hyperinsulinemia are not only experienced systemically, but within the TME, a finding that connects obesogenic circadian rhythm disruption to tumor clock genes and suggests that TRF, a low-cost, non-pharmacological intervention could be used as an adjuvant to improve cancer therapies. Only minor comments are noted below.

Specific Points:

1. Page 1, line 4: Author affiliation for Mehakdeep Kau.
2. Page 4, lines 74-75: "Not surprisingly, weight loss in obese individuals by caloric restriction or fasting leads to beneficial effects on metabolism and cancer growth in both mice and humans." Insert modifier to clarify that CR or fasting leads to beneficial effects at reducing cancer growth in mice and humans.
3. Page 4, lines 80-82: "Epidemiological studies have shown that prolonged nightly fasting (period between last meal at night and first meal of the day) is associated with reduced breast cancer risk and recurrence 18, 23-26." Since reference 24 discusses meal timing in mouse models, broaden statement to say, "Epidemiological and preclinical models show that...".
4. Page 4, lines 87-88: "Promising benefits have been observed in pilot studies in humans but appropriately powered, randomized controlled trials are lacking." Expand to include disease state of study subjects and what specific metabolic or health-related promising benefits were observed in these pilot studies.
5. Page 4, lines 88-90: "Importantly, TRF in both sexes of obese mice improves overall metabolic health and reduces insulin resistance without weight loss and a small (n=8) all-male study suggests this is also true in humans 29, 31, 34." Translational relevance and potential impact of sex should be discussed in greater detail.
6. Page 5, lines 97-100: Rephrase to include a hypothesis statement.
7. Page 6, line 119: Citation for "Similar to what we have reported in female OVX mice..."
8. Figure 1 and supplemental figures 1 and 2. Were the local reductions in pro-inflammatory mediators in the liver and mammary adipose tissue in response to TRF observed systemically?
9. Page 7, lines 152-154. Citation 35, Bao, et al. states that, "Py230 cells only give rise to hormone receptor-positive tumors when injected orthotopically at relatively low cell densities (1 x 10⁴ cells or less), suggesting that the formation of hormone receptor-positive tumors may result from paracrine signaling between normal mammary cells and tumor cells". However, page 16, line 361 states that the Py230 cells were injected at 1 x 10⁵ cells bilaterally into the #3 and #4 mammary fat pads. Did the authors confirm that their method of injection did in fact result in estrogen and progesterone positive luminal-A tumors?

10. Page 7, lines 157-158 and page 8, lines 160-161: It appears that Figure 2C is the same as supplemental figure 4C and that Figure 4F is the same as supplemental figure 4E?
11. Page 8, lines 161-165: Perhaps a subtle effect, but were the lean mice that underwent the TRF age-matched to the typical age of the HFD-TRF mice? Would be interesting to know if the extra 10 weeks of standard chow diet could induce some level of dysregulation that could be improved by TRF.
12. Page 8, lines 168-171: The figure legend states that Figure D-G is from OVX mice, but this distinction could be added here in text.
13. Page 8, lines 173-175: The phosphorylation of these signaling mediators can occur through insulin-dependent and insulin-independent mechanisms, so perhaps a broader statement should be used to incorporate the potential for other growth factors, etc. to impact these signaling pathways.
14. Page 8, line 178: Justification for a three-week TRF lead-in period for the Py230 vs. a six-week TRF lead-in for the EO771 prior to tumor?
15. Page 9, lines 186-189: Tail vein injection bypasses several important steps in the metastatic process. Were lung metastases quantified in the orthotopically injected mice presented in Fig 3A?
16. Page 9, lines 200-201: Without showing body weights in chow-fed animals, it is difficult to know that the TRF maintained “normal” body weight. Why were chow-fed controls not included in this transgenic model, but were in the two orthotopic models?
17. Page 9, lines 201-202: “Importantly, as in the other models, the TRF group consumed the same number of calories as the AL group.” Is this data not shown?

Reviewer #3 (Remarks to the Author):

This is an interesting paper that provides new data that supports concepts already in the literature.

My major concern is not that data are flawed, but that conclusions are closely related to prior work, although of course the precise model described is novel.

minor points - some of the introductory references concerning obesity are old rather than current; the complex relationships between menopausal status, obesity and breast cancer are over-simplified.

Also, its obvious that in patients with advanced metastatic breast cancer and cachexia, there is substantial reduction in caloric intake but cancers are behaving aggressively - how does that relate to the hypothesis and data reported -- perhaps the findings are not generalizable to all stages of breast cancer.

For a review of prior related work that makes the reported results plausible but not highly original, see NEJM 381: 2541-51.

Reviewer #4 (Remarks to the Author):

The manuscript by Das et al. reports a series of remarkable experiments that document the effect of time restricted feeding (TRF), as compared to ad libitum (AL), on breast tumorigenesis in several mouse models. The authors report that hyperinsulinemia in AL versus improved glucose tolerance with TRF underlies the increased tumorigenesis with the inference that insulin itself is mitogenic to breast cancer cells. They document that exogenous insulin could block the effects of TRF and an anti-diabetic drug could curb the effects of AL on increasing tumorigenesis. Further, they show using the single cell line Py230 that insulin behaves as a mitogen in vitro. This observation is consistent with the findings that TRF treated tumors had diminished Ki67. Remarkably, the authors then report that TRF increases synchronicity of tumor circadian clock through a 4h sampling time course for circadian gene expression. This phenomenon is diametrically opposite of the effects of exogenous insulin (hyperinsulinemia) which caused clock dysrhythmia. Collectively, the authors report foundational studies providing a mechanistic framework for the effects of TRF and its seemingly positive correlation with human breast cancer tumorigenesis reported from UCSD. There are important points that the authors should address, at the minimum – in the discussion.

1. The thesis that insulin is mitogenic for breast cancer cells and underlies the key mechanism is supported by the study of one cell line Py230 in vitro. Note that while this is documented (and expected to some extent, ie, not novel), it is notable that Py230 is ER+ and PR+ and the authors did not measure the effects of estrogen or progesterone on these cells in reference to the potential changes of these hormones in vivo in response to TRF (which was not addressed).
2. The corresponding changes in Ki67 and vascularity in situ (by histological studies) are construed to connect to the in vitro findings. The authors have not directly addressed the potential non-cell-intrinsic effects of TRF (or conversely, the systemic effects of hyperinsulinemia); for example, on the immune system.

3. The authors did not state which cell model was used for the circadian studies (Fig 7) which demonstrate that exogenous insulin caused clock dysrhythmia in tumors. Confusingly, the authors report in Supplemental Fig 7 that insulin could synchronize Py230 and E0771 clocks in vitro. Closer examination of the in vitro experiment indicates that the time course was over 8 hours after 10ng insulin stimulation. The time course is insufficient to suggest that “insulin caused rhythmic expression of many clock genes.” Ideally, a every 4h sampling should be done over 48h. Hence, the data in Supplemental Fig 7 are uninformative, particularly since the time course for control was not shown. Further the behavior of some of the transcripts suggests a possible artifact of response to insulin itself (ie the mitogenic effect) and that Bmal1 expression was not anti-phase to Rev-erb-alpha (which would be expected, but this short time 8h course is uninformative). Given these limitations, the authors have not established whether these tumor cell lines have functioning clocks and that these clocks were synchronized by TRF in vivo. Also, it is possible that in the instance that these cell lines have disrupted intrinsic clocks, the in vivo milieu could cause rhythmic expression of clock genes in vivo due to other systemic factors. This entire section needs to be revised.

Response to the reviewer's comments

We thank the reviewers' for their positive comments. We apologize that the revision of the manuscript took so long but we performed two completely new TRF experiments that took 20 weeks, and we were faced with lab shutdown due to the pandemic. The changes to the manuscript are high-lighted in yellow.

Reviewer #1 (Remarks to the Author):

Summary to Author: The manuscript "Time-restricted feeding normalizes hyperinsulinemia to inhibit breast cancer in obese postmenopausal mouse models" by Das et al. includes the data presented at ENDO 2019 meeting (J Endocr Soc. 2019 Apr 15; 3 Suppl 1) published online. It is well established that calorie restriction (CR) reduces the progression of tumors in various animal models, although the efficacy of CR may be limited to certain cancers. This study narrows in on post-menopausal breast cancer initiation and progression in the context of time-restricted feeding (TRF), a time (but not caloric)-restricted approach to energy intake which improves metabolism in a number of tissues and systemically under conditions of high fat diet feeding. The manuscript is well written and addresses a clinically important issue. By using HFD-fed postmenopausal mouse models, the authors show that TRF reduces obesity and improves metabolic dysregulation in obese reduces Py230 mammary tumor growth, attenuates E0771 mammary tumor growth and metastasis to lung, and inhibits tumor initiation and growth in obese OVX-PyMT transgenic mice concomitantly with normalizing endogenous circadian rhythms in breast tumors. They provide evidence suggesting that TRF acts through modulating hyperinsulinemia. Implication of insulin is expected, as obesity-related insulin resistance is a well-known risk factor for breast cancer development in post-menopausal women. However, injected insulin was not found to affect cancer risk in clinical trials, and it is clear how this study adds to resolving this controversy. The overall effects of TRF on tumor growth and metastasis are convincing. However, there is a lack of sufficient detail related to the mechanisms involved. The lack of detailed attention to zeitgeber times, as well as the absence of comparison to normal breast tissue, limit interpretation in a number of areas. The links between insulin signaling and clock proteins (BMAL1 specifically) should be addressed both experimentally and in the text.

Response: The reviewer is correct that insulin injection is not found to affect cancer risk, however the insulin injections only provide a transient increase and no studies have investigated the effect of chronic hyperinsulinemia on cancer risk. A number of high-affinity insulin analogs were tested by pharmaceutical companies but not taken into the clinic due to increased cancer incidence in preclinical models. So, there is some unpublished evidence that insulin injections can increase cancer risk. Also we have greatly increased the analysis of circadian effects in the normal tissue and the tumor as requested by the reviewer, and also included assessments of AKT signaling and BMAL1 phosphorylation. (page 13-15)

Specific comments:

Comment 1. The authors should address whether OVX or VCD treatment affects overall rhythmicity of the animal, considering the focus of the study is on time-restricted feeding. How diurnal behavior is affected with and without TRF is not discussed, nor how it compares between the two different models.

Response: A number of groups have shown that diet-induced obesity disturbs diurnal behaviors in male mice, for example Hatori et al. in intact male mice fed HFD (PMID: 22608008), and these disturbances are corrected by TRF. For females, the data is sparse. In our previous study (Chung et al., Metabolism, 2016, PMID: 27832862) in the obese OVX mouse model, we documented the effect of TRF on diurnal

behaviors. In a CLAMS study we measured food intake, water intake, X ambulatory (e.g., walking/running movement in the X-axis plane) and Z activity (e.g., reaching/rearing in the Z-axis plane) over 24 h. Our results indicate that TRF positively improves the diurnal feeding and locomotor behavior compared to ad libitum feeding. There is some older data that gonadal steroids alter sleep rhythms in rats which could affect metabolism (e.g. PMID: 6101987, 23843233). This mirrors the known circadian sleep disturbances seen in menopausal women (PMID: 27585541). A recent 2019 publication from the Pendergast lab showed that estrogen regulates circadian rhythms to protect against diet-induced obesity (PMID: 31689145) and ovariectomy results in disrupted circadian rhythms when placed on HFD. In particular, ovariectomy did not alter circadian activity in female mice on LFD but did make them more susceptible to HFD effects. Similar results were found in OVX rats (PMID: 31585360). Any disturbance of circadian rhythms in the OVX model is likely appropriate for post-menopausal women who have similar disturbances. Circadian activity has not been studied in the VCD menopause model, but we expect any changes to be less than the OVX model as gonadal steroids are not eliminated. Comments to this effect have been included in the introduction to the OVX model. (Page 4, line 70-72, page 6 line 123-125).

Comment 2. Related to point #1, can the authors provide the total food intake presented in Figure 1 into day vs. night energy intake? Though ad libitum-applied HFD has been proposed to reduce rhythmicity of energy intake, this is somewhat controversial. It would be nice to see the extent to which the nice tumor reduction effects have anything to do with restored rhythms in energy intake.

Response: We did not measure day vs night food intake in this study, as we had done so in our previous paper on the obese OVX mouse model (Chung et al., *Metabolism*, 2016, PMID: 27832862). We published that the **total** food intake per day was the same for TRF and ad libitum HFD. When we looked at diurnal differences in behavior, the AL HFD group showed a small but statistically significant difference between day and night consumption in a CLAMS study, whereas the NC showed a more pronounced diurnal difference (adapted from Chung et al). Obviously the TRF group has an enforced rhythm. We also saw differences in locomotor activity between the two groups.

Our results are consistent with the view that HFD ad libitum feeding reduces diurnal differences in feeding behavior but does not eliminate them.

Similar observations on diurnal food intake has also been reported by Hatori et al. in male mice fed HFD (*Cell Metabolism*, 2012, PMID: 22608008). Mice eat the same amount of food so differences in tumor growth cannot be attributed to caloric intake. We show that tumor cells cannot use ketone bodies as an energy, so the prolonged fasting, which causes glucose to drop and ketones to rise, might be cytostatic in that the cells are prevented from proliferating during the fast. This needs testing and we are planning ways to address this idea, but it lies beyond the scope of this paper.

Comment 3. The authors do not discuss in sufficient depth the relationship between insulin signaling and the circadian clock. TRF has been reported to function in part by restoring rhythmicity in various organs. BMAL1 itself is downstream of IR, getting phosphorylated directly by AKT and GSK3B. The results of Figure 2K suggest that BMAL1 stability would be altered under TRF vs. AL and NC. Is this the case? I think that additional analysis of circadian molecules in the breast stroma and tumor tissue would indicate whether TRF functions within the tumor to affect core clock proteins vs. is functioning in a largely clock-

independent manner. Rather than simply showing gene expression in Figure 7, BMAL1 protein should be directly analyzed, particularly its phosphorylation as antibodies are available and widely used for such analysis. This is a critical point, as BMAL1 has emerged as a key molecule in several tumor types, with under- or over-expression having different effects depending on the tumor type.

Response: As per the reviewer's suggestion we incorporated the relationship between insulin and the circadian clock in the discussion section (**Page 18, Line 416-423**). Further, we have included extensive new data on circadian gene and protein expression in the mammary fat tissue (breast stroma), liver and tumor tissue from the NC, AL, TRF groups of mice harvested every 4 hours over 24 hours (ZT 0, 4, 8, 12, 16, 20) to elucidate whether TRF functions within the tumor to affect core clock proteins (**see new Figure 8, Figure 9, Supplemental figure 7, 8, 9 and 10, Supplementary Table 1 and 2**). Our results clearly show that specific clock proteins are affected in a positive way by TRF to normalize rhythms. The text has been edited accordingly (**Page 13, line 283-338 and Page 18, Line 399-430**). As TRF alters AKT phosphorylation (as the reviewer points out) and Bmal1 is phosphorylated by AKT, we also measured BMAL1Ser42 phosphorylation and showed that it is circadian with TRF causing robust pBMAL1 rhythms in the liver and MFP consistent with the reviewer's suggestion, but not in the tumor where AL seemed to give the strongest rhythms (**see new Figure 9 and Supplemental Figures 8 and 9**).

Comment 4. The authors discuss Cidea and Cidec as being elevated in ad libitum-fed mice compared to TRF. These are highly dynamic molecules in the context of high fat diet feeding in several tissues. The authors do not state at which zeitgeber time these are being analyzed.

Response: All the mice were sacrificed and tissues collected at ZT4-6 and the details have been incorporated in the revised text (**Page 21 line 483 and legend to supplemental Figure 2**).

Comment 5. Figure 7 addresses the rhythmicity of clock genes in tumors of normal chow, ad-libitum HFD-fed mice, and TRF HFD-fed mice. What is very confusing about these data is that Per and Bmal1 are in the same phase. (In fact, all of the clock genes shown are in the same phase, suggesting the possibility of some normalization artifact.) Though Per1 and Bmal1 are generally anti-phasic in expression, the authors state that TRF normalized rhythms to normal in the tumor tissues. These are in no way "normal"- at least not compared to healthy cells. Here, it would be helpful to have the non-tumor tissue as a comparison. As presented, these data are very difficult to interpret what is going on with the clock in the various conditions. To the authors' credit, this issue is finally discussed briefly in the discussion section. I think some experimental approach (i.e. comparing tumor to normal breast stroma) would be a better way to address this strange result. In addition, amplitude of oscillation could be compared between tumor and non-tumor. Finally, it has been reported that diets rich in saturated and omega 6 fatty acids are pro-inflammatory and increase breast cancer risk, while diets rich in omega 3 fatty acids are anti-inflammatory and decrease it. It would be interesting to test if/how the link of dietary intake with cancer initiation and progression is affected by diet composition, and specifically by the types of fatty acids in HFD.

Response: At the reviewer's suggestion we repeated the circadian experiment and collected tissues under light and dark conditions every 4 hours over 24 hours (ZT 0, 4, 8, 12, 16, 20) without altering the feeding schedule. We measured core circadian gene and protein expression in the mammary fat tissue (breast stroma), liver and tumor tissue from the NC, AL, TRF groups of mice. Data are shown in new **Figure 8, Figure 9, Supplemental figure 7, 8, 9 and 10, Supplementary Table 1 and 2**. We analyzed the data two ways. Firstly the 24 h data were analyzed using the RAIN software (Rhythmicity Analysis Incorporating Nonparametric methods), then the data were modeled using a cosine function in PRISM. Then data were considered circadian if RAIN indicated significant rhythms and if the cosine function was a better fit than

a straight line in PRISM. For ease of visualization only, the data were doubled and plotted over 48 h as is commonly done in circadian activity plots, but we would like to stress that the statistical significance is based on the 24 h data only. The text has been edited accordingly (**Page 13 line 283-338 and Page 18, line 399-430**). Details regarding rhythmicity, phase and amplitude changes in different genes in tumor vs non-tumor tissue were also compared and incorporated in revised text (**Page 13 line 283-338**). These results now clearly indicate the important role TRF plays in regulating circadian gene and protein expression.

We agree that the role of different diets fed using a TRF protocol would be interesting to study, as we have previously published that diets rich in omega-3 fatty acids can inhibit tumor growth like TRF. However, those studies are beyond the scope of this paper.

Reviewer#2 (Remarks to the Author):

Summary to Author: This timely, novel, and well-designed study provides intricate detail into a complex area of research. Strengths include characterizing the metabolic effects of TRF in both tumor-free and tumor-bearing surgical and chemically induced ovariectomized mice, and the use of multiple mammary tumor models (orthotopic and transgenic), numerous techniques and “break and fix” methods, and time course outcomes to identify circadian rhythm genes altered with HFD and responsive to TRF within the tumor microenvironment. In total, the data presented in this manuscript paints a very thorough picture and provides new insights into how a TRF paradigm can normalize high-fat diet-induced insulin levels to reduce tumor outgrowth in preclinical models of postmenopausal breast cancer. Interestingly, the TRF benefits on normalizing hyperinsulinemia are not only experienced systemically, but within the TME, a finding that connects obesogenic circadian rhythm disruption to tumor clock genes and suggests that TRF, a low-cost, non-pharmacological intervention could be used as an adjuvant to improve cancer therapies. Only minor comments are noted below.

Specific comments:

Comment 1. Page 1, line 4: Author affiliation for Mehakdeep Kaur.

Response: Author affiliation for Mehakdeep Kaur has been edited in the revised MS.

Comment 2. Page 4, lines 74-75: “Not surprisingly, weight loss in obese individuals by caloric restriction or fasting leads to beneficial effects on metabolism and cancer growth in both mice and humans.” Insert modifier to clarify that CR or fasting leads to beneficial effects at reducing cancer growth in mice and humans.

Response: As per the reviewer’s suggestion, a modifier has been inserted in the revised text (**page 4, line 83-84**) to clarify that CR or fasting leads to beneficial effects at reducing cancer growth in mice and humans.

Comment 3. Page 4, lines 80-82: “Epidemiological studies have shown that prolonged nightly fasting (period between last meal at night and first meal of the day) is associated with reduced breast cancer risk and recurrence 18, 23-26.” Since reference 24 discusses meal timing in mouse models, broaden statement to say, “Epidemiological and preclinical models show that...”.

Response: As per the reviewer’s suggestion we edited the above statement in the revised MS (**Page 4, line 89-91**).

Comment 4. Page 4, lines 87-88: “Promising benefits have been observed in pilot studies in humans but appropriately powered, randomized controlled trials are lacking.” Expand to include disease state of study subjects and what specific metabolic or health-related promising benefits were observed in these pilot studies.

Response: As per the reviewer’s suggestion we expanded the human pilot studies to include the disease state of the subject and metabolic benefits observed (**Page 5, 96-105**).

Comment 5. Page 4, lines 88-90: “Importantly, TRF in both sexes of obese mice improves overall metabolic health and reduces insulin resistance without weight loss and a small (n=8) all-male study suggests this is also true in humans 29, 31, 34.” Translational relevance and potential impact of sex should be discussed in greater detail.

Response: In the discussion section of the revised manuscript, we have included further discussion of the translational relevance of TRF in relation to hyperinsulinemia and breast cancer (**Page 17, line 379-398**) as well as circadian rhythms and breast cancer (**Page 18, line 399-430**). Because we are interested in breast cancer, we tested TRF in female mice in the relevant postmenopausal models rather than studying breast cancer in male mice that are commonly used to study diet-induced obesity. Unfortunately, there is a common misconception that female mice do not become obese and insulin resistant on HFD that has led to the predominance of male studies. However, we and others have published many papers showing female mice do indeed become obese and insulin-resistant, but it takes slightly longer. As suggested, we discussed the importance of sex difference and potential implications in the revised MS (**Page 17, Line 374-378**). We also have discussed the importance of sex differences in our previous manuscript (Chung et al, *Metabolism*, 2016, PMID: 27832862).

Comment 6. Page 5, lines 97-100: Rephrase to include a hypothesis statement.

Response: As suggested, we rephrased the sentence to include a hypothesis statement in our revised MS (**Page 5, line 1012-114**)

Comment 7. Page 6, line 119: Citation for “Similar to what we have reported in female OVX mice...”

Response: As suggested, we have incorporated the citation for “Similar to what we have reported in female OVX mice...” in our revised MS (**Page 7, line 137**).

Comment 8. Figure 1 and supplemental figures 1 and 2. Were the local reductions in pro-inflammatory mediators in the liver and mammary adipose tissue in response to TRF observed systemically?

Response: We did not measure circulating cytokines in these studies. However, recent studies by Sundaram et al. in breast cancer and by Yan et al. in lung cancer mouse models have demonstrated that TRF reduces systemic pro-inflammatory cytokines including ccl2 compared to AL HFD (S. Sundaram et al., PMID: 30442235, L. Yan et al. PMID: 30952713). Furthermore, T. Moro et al. showed in healthy resistance-trained males that TRF reduces plasma interleukin-6, interleukin-1 β , tumor necrosis factor α compared to standard diet group (Moro et al, *J Transl Med.*, 2016, <https://doi.org/10.1186/s12967-016-1044-0>). So there may well be reductions in circulating cytokines, but we did not measure them as we were focused on the hyperinsulinemia as the driver of tumor growth. We had previously shown that reducing inflammation did not reduce tumor growth in this obese post-menopausal model so did not pursue it here (Chung et al. PMID: 25220417)

Comment 9. Page 7, lines 152-154. Citation 35, Bao, et al. states that, “Py230 cells only give rise to hormone receptor-positive tumors when injected orthotopically at relatively low cell densities (1 x 10⁴ cells or less), suggesting that the formation of hormone receptor-positive tumors may result from paracrine signaling between normal mammary cells and tumor cells”. However, page 16, line 361 states that the Py230 cells were injected at 1 x 10⁵ cells bilaterally into the #3 and #4 mammary fat pads. Did the authors confirm that their method of injection did in fact result in estrogen and progesterone positive luminal-A tumors?

Response: We performed IHC of Py230 tumors (100,000 cells injected) from OVX mice using anti-estrogen receptor antibody (D12, 1:100 Santa Cruz) or anti-progesterone receptor antibody (ab63605, 1:500 Abcam) to confirm that at this cell density we were able to produce estrogen and progesterone positive luminal-A tumors. The IHC images are included in Supplemental figure 6. We also showed that estradiol and progesterone stimulated proliferation of Py230 cells in vitro supporting the idea that they express ER and PR (Supplemental figure 6). We also edited the text accordingly (**Page 12, line 254-258**).

Comment 10. Page 7, lines 157-158 and page 8, lines 160-161: It appears that Figure 2C is the same as supplemental figure 4C and that Figure 2F is the same as supplemental figure 4E?

Response: Thank you for noticing our error. We have included the correct figure 2C and 2F in our revised MS

Comment 11. Page 8, lines 161-165: Perhaps a subtle effect, but were the lean mice that underwent the TRF age-matched to the typical age of the HFD-TRF mice? Would be interesting to know if the extra 10 weeks of standard chow diet could induce some level of dysregulation that could be improved by TRF.

Response: The lean mice that underwent the TRF were age-matched to the typical age of the HFD-TRF mice. The text has been revised accordingly (**Page 21, line 478-480**)

Comment 12. Page 8, lines 168-171: The figure legend states that Figure D-G is from OVX mice, but this distinction could be added here in text.

Response: As suggested we edited the text accordingly in our revised MS (**Page 9, line 186-188**)

Comment 13. Page 8, lines 173-175: The phosphorylation of these signaling mediators can occur through insulin-dependent and insulin-independent mechanisms, so perhaps a broader statement should be used to incorporate the potential for other growth factors, etc. to impact these signaling pathways.

Response: As suggested, we edited the text accordingly in our revised MS (**Page 9, line 191-193**)

Comment 14. Page 8, line 178: Justification for a three-week TRF lead-in period for the Py230 vs. a six-week TRF lead-in for the EO771 prior to tumor?

Response: As the EO771 cells are very aggressive cells compared to Py230 cells, the tumor growth is much faster. So, in order to have the same 10 weeks of TRF intervention we had a 6 weeks TRF lead in for the EO771 prior to tumor.

Comment 15. Page 9, lines 186-189: Tail vein injection bypasses several important steps in the

metastatic process. Were lung metastases quantified in the orthotopically injected mice presented in Fig 3A?

Response: In E0771 orthotopically injected tumor experiment we had to euthanize the mice as per our IACUC criteria for euthanizing the mice with particular tumor size. The only way to see metastasis in this model is to remove the primary tumor, which then gives enough time for metastases to be observed. We did not do this as we wanted to harvest liver and mammary fat pad at the same time as the tumors.

Comment 16. Page 9, lines 200-201: Without showing body weights in chow-fed animals, it is difficult to know that the TRF maintained “normal” body weight. Why were chow-fed controls not included in this transgenic model, but were in the two orthotopic models?

Response: We edited the text in our revised MS to give a better understanding on TRF effect on body weight (Page 10, Line 218-219). In the two-orthotopic models we have clearly demonstrated that TRF is effectively reducing the tumor growth compared to HFD-ad libitum and the tumor growth is similar to NC control group (Figure 2 and 3). TRF also reduced the tumor growth compared to HFD-ad libitum in pre-existing tumor (Figure 5) and the tumor growth is similar to NC control group. In the PyMT transgenic mouse model, we could not make the animals obese prior to tumor formation, as we had in the other experiments, as the tumors appear before the mice become obese. So, we put them on TRF HFD from the start. Satchin Panda’s lab has shown that TRF feeding of a HFD to lean mice prevents weight gain, and the mice maintain their normal bodyweight. Our primary aim was to validate the effect of TRF on tumor initiation in a spontaneous model, so we did not include a NC group as the TRF group was in effect the normal weight control group comparator. Previously we and others (PMID: 8530414, PMID: 26992445, PMID: 17206489) have published the growth characteristics of tumors in the PyMT mice on normal chow. The PyMT phenotype depends on strain background. Tumors appear earlier and grow faster on the FVB background than the C57BL6J background.

Comment 17. Page 9, lines 201-202: “Importantly, as in the other models, the TRF group consumed the same number of calories as the AL group.” Is this data not shown?

Response: As suggested, we incorporated the data in Figure 4L.

Reviewer #3 (Remarks to the Author):

This is an interesting paper that provides new data that supports concepts already in the literature. My major concern is not that data are flawed, but that conclusions are closely related to prior work, although of course the precise model described is novel.

Minor points

Comment 1: Some of the introductory references concerning obesity are old rather than current; the complex relationships between menopausal status, obesity and breast cancer are over-simplified.

Response: As suggested, we replaced the older reference in the introductory section concerning obesity with current reference. We also did the necessary edits in our revised MS to represent the complex relationships between menopausal status, obesity and breast cancer (**Page3, line 52-61**).

Comment 2. Also, its obvious that in patients with advanced metastatic breast cancer and cachexia, there is substantial reduction in caloric intake but cancers are behaving aggressively - how does that relate to the hypothesis and data reported -- perhaps the findings are not generalizable to all stages of breast

cancer. For a review of prior related work that makes the reported results plausible but not highly original, see NEJM 381: 2541-51.

Response: The reviewer points out that in patients with advanced metastatic breast cancer and cachexia, there is often a reduction in caloric intake, while the cancers are growing aggressively. In these cases, we doubt that timed eating would have a direct effect on tumor growth, although there is data that fasting can increase the efficiency of chemotherapy. The reduced caloric intake in these patients is likely due to treatment side-effects. This is different from the daily fasting and feeding rhythm that might be playing a role in improved metabolism and thereby tumor suppression. If TRF has an effect in humans, I would expect to see the greatest benefit in early stage disease. If we can show that TRF synergizes with chemotherapy (we are currently performing those experiments in mice) then TRF may have applicability in more advanced disease. The review that the reviewer cites is from Mark Mattson who studies intermittent fasting. In our opinion, IF is different from TRF as IF involves caloric restriction whereas TRF does not. So, we feel that TRF should not be considered a form of IF, although the media and other investigators do not make that distinction. TRF is circadian driven but IF is not.

Reviewer #4 (Remarks to the Author):

The manuscript by Das et al. reports a series of remarkable experiments that document the effect of time restricted feeding (TRF), as compared to ad libitum (AL), on breast tumorigenesis in several mouse models. The authors report that hyperinsulinemia in AL versus improved glucose tolerance with TRF underlies the increased tumorigenesis with the inference that insulin itself is mitogenic to breast cancer cells. They document that exogenous insulin could block the effects of TRF and an anti-diabetic drug could curb the effects of AL on increasing tumorigenesis. Further, they show using the single cell line Py230 that insulin behaves as a mitogen in vitro. This observation is consistent with the findings that TRF treated tumors had diminished Ki67. Remarkably, the authors then report that TRF increases synchronicity of tumor circadian clock through a 4h sampling time course for circadian gene expression. This phenomenon is diametrically opposite of the effects of exogenous insulin (hyperinsulinemia) which caused clock dysrhythmia. Collectively, the authors report foundational studies providing a mechanistic framework for the effects of TRF and its seemingly positive correlation with human breast cancer tumorigenesis reported from UCSD. There are important points that the authors should address, at the minimum – in the discussion.

Comment 1. The thesis that insulin is mitogenic for breast cancer cells and underlies the key mechanism is supported by the study of one cell line Py230 in vitro. Note that while this is documented (and expected to some extent, ie, not novel), it is notable that Py230 is ER+ and PR+ and the authors did not measure the effects of estrogen or progesterone on these cells in reference to the potential changes of these hormones in vivo in response to TRF (which was not addressed).

Response: As per the reviewer's suggestion we first measured progesterone and estradiol level in response to time restricted feeding of HFD (TRF) (Fig 2), which indicates that there is no significant difference in TRF vs Ad libitum HFD group. Further we observed that in VCD mice, the progesterone level is little higher than the OVX mice (Figure 1B of the manuscript), however the effect of TRF on tumor growth is similar in both of the mouse model. This indicates that TRF is effective irrespective of the Progesterone level. We do know that Py230 tumors are estrogen sensitive in vivo as tumors grow much faster in intact female mice than OVX mice. Obesity does not increase tumor growth in intact mice consistent with the premenopausal data in women. So TRF is unlikely to be effective as tumor growth is driven by estrogen not insulin. We also studied the effect of estrogen and progesterone on Py230 cells

(Supplemental Figure 6). The result demonstrated that both progesterone and estradiol increase the proliferation of py230 cells dose dependently.

Fig. Analysis of progesterone and Estradiol hormone level in plasma of Py230 tumor bearing ovariectomized (OVX) mice on time restricted feeding (TRF) or ad libitum feeding (AL) of HFD.

Comment 2. The corresponding changes in Ki67 and vascularity in situ (by histological studies) are construed to connect to the in vitro findings. The authors have not directly addressed the potential non-cell-intrinsic effects of TRF (or conversely, the systemic effects of hyperinsulinemia); for example, on the immune system.

Response: Previously we found that obesity-driven tumor growth in the orthotopic Py230 model is not driven by inflammation. Here we find that hyperinsulinemia is driving growth. We can demonstrate a mitogenic effect of insulin on the Py230 cells themselves, but it does not rule out an indirect effect as the reviewer suggests. It is possible that the hyperinsulinemia alters the local micro-environment to make it more receptive to tumor initiation. These would be very interesting to study but are beyond the scope of this manuscript.

Comment 3. The authors did not state which cell model was used for the circadian studies (Fig 7) which demonstrate that exogenous insulin caused clock dysrhythmia in tumors. Confusingly, the authors report in Supplemental Fig 7 that insulin could synchronize Py230 and E0771 clocks in vitro. Closer examination of the in vitro experiment indicates that the time course was over 8 hours after 10ng insulin stimulation. The time course is insufficient to suggest that “insulin caused rhythmic expression of many clock genes.” Ideally, a every 4h sampling should be done over 48h. Hence, the data in Supplemental Fig 7 are uninformative, particularly since the time course for control was not shown. Further the behavior of some of the transcripts suggests a possible artifact of response to insulin itself (i.e the mitogenic effect) and that Bmal1 expression was not anti-phase to Rev-erb-alpha (which would be expected, but this short time 8h course is uninformative). Given these limitations, the authors have not established whether these tumor cell lines have functioning clocks and that these clocks were synchronized by TRF in vivo. Also, it is possible that in the instance that these cell lines have disrupted intrinsic clocks, the in vivo milieu could cause rhythmic expression of clock genes in vivo due to other systemic factors. This entire section needs to be revised.

Response: We have completely redone the circadian experiment as mentioned in our response to reviewer 1. The *in vivo* circadian study was done in Py230 tumor model and detail of the protocol is described in the Method section (**Page 21, line 463-467, 482-488**). We studied different circadian genes and their protein expression in the mammary fat tissue (breast stroma), liver and tumor tissue from the NC, AL, TRF groups of mice. Tissues were harvested every 4 hours over 24 hours (ZT 0, 4, 8, 12, 16, 20) to elucidate TRF mechanisms (**Figure 8, Figure 9, Supplemental figure 7, 8, 9, Supplementary Table 1 and 2**). Furthermore, we performed an *in vitro* circadian experiment using serum shock to stimulate rhythms in Py230 cells to show that the cells have an intrinsic clock (**Figure 8E, Supplemental figure 10, Page 29, line 661-670**). Interestingly the tumors have a clock, but the phase of some genes does not match the liver and MFP suggesting that the clock may not function the same way. The text has been edited accordingly (**Page 13, line 283-338 and Page 18, line 399-430**). Details regarding rhythmicity, phase and amplitude changes in different genes in tumor vs non-tumor tissue were also compared and

incorporated in revised text.

REVIEWERS' COMMENTS

Reviewer #1 (Remarks to the Author):

The authors have adequately responded to all of my prior criticisms and have gone to a significant amount of effort to improve their manuscript in several ways.

Specifically, additional text has been added to the manuscript concerning the possible changes in diurnal patterns in their VCD menopause model. (Though the authors do not report light vs. dark energy intake from this experiment, in their response they refer to earlier food intake measurements from their own group that support patterns of eating behavior with TRF.) Secondly, the new experiment performed to analyze gene expression in the different feeding conditions and tumor vs. non-tumor is a great improvement to the manuscript. The circadian genes now show the expected phase relationships and the results of TRF in the tumor are quite striking. In addition, the BMAL1 phosphorylation data underscore the relationship between insulin signaling the core clock, a property of the clock not well addressed in the first submission.

Reviewer #2 (Remarks to the Author):

The authors have successfully addressed all my concerns and improved this already very thorough and important translational study.

Reviewer #3 (Remarks to the Author):

The paper is improved but I still have reservations regarding its suitability for Nature Communications. The technical details are fine, but let's look at the 'big picture' and novelty. Here is a paragraph from the discussion:

"In summary we found that TRF normalizes hyperinsulinemia, restores circadian rhythms, and inhibits tumor growth in models of post-menopausal obesity-driven breast cancer. These findings provide a molecular rationale for the adoption of TRF as a life-style change in humans to reduce insulin resistance and normalize circadian rhythms. Thus, TRF will likely be an effective non-pharmacological intervention to prevent or inhibit the progression of human breast

cancer, and may also provide adjunctive benefit when used in combination with standard chemotherapy"

First, findings in murine models often do not lead to similar findings in humans, "will likely be effective" is too strong a statement - 'clinical trials to determine if the model has relevance to human cancer are justified' might be better.

Second, the authors appropriately cite prior literature (their refs 42,43,36,38) that TRF lowers insulin in mice and humans, so recapitulating this is nice, but certainly not novel, and "we found that that TRF normalizes hyperinsulinemia" might be replaced by "we confirmed and extended prior reports that TRF normalizes hyperinsulinemia" .

Finally, the notion that adverse effects of obesity on post menopausal breast cancer are mediated at least in part by insulin is "old news" and is in the literature in many (>3000 on pubmed) papers, some cited by the authors, but many important ones and reviews not cited, including relevant Mendelian randomization studies.

I feel the work deserves publication somewhere, but in view of the above if it has sufficient novelty for Nature Communications should be decided by the editors.

Certainly, if it is accepted, the discussion should be altered to be more modest and show how building on prior knowledge related to insulin and cancer (perhaps cite some of the classic and/or newer comprehensive reviews to this large field - eg: Rachel J Perry 1, Gerald I Shulman. Mechanistic Links between Obesity, Insulin, and Cancer. Trends Cancer 2020 Feb;6(2)75-78; Pollak M. The insulin and insulin-like growth factor receptor family in neoplasia: an update.

Nat Rev Cancer. 2012 Feb 16;12(3):159-69) and also building on prior knowledge appropriately cited by authors regarding insulin-lowering effects of TRF, it was shown that TRF has an inhibitory effect on a murine model of breast cancer. Other ways of lowering insulin have previously been shown to have tumor inhibitory effects on various kinds of cancer in mouse models (dozens of papers, a recent example: Uncoupling Hepatic Oxidative Phosphorylation Reduces Tumor Growth in Two Murine Models of Colon Cancer. Wang Y et al Cell Rep. 2018 Jul 3;24(1):47-55.) - thus a strength of the paper is not the concept of insulin reduction, but rather the possibility that a simple diet manipulation might be the most practical way to achieve this clinically. Nevertheless, it remains completely unclear if the magnitude of the insulin reduction achieved this way would be sufficient to have a clinical impact.

The editor should have an opinion if the novelty is sufficient - I think its borderline; but if he/she wants to publish the paper, I suggest a further revision to better position the observations in the context of the prior literature, and not to go beyond the data with respect to speculation regarding clinical relevance.

Reviewer #4 (Remarks to the Author):

The authors have substantively addressed issues I raised as well as other issues.

Response to Reviewer's comments

We have made a number of changes to the manuscript in response to Reviewer 3's comments. We have toned down the language as requested and have included more references to some of the previous work. The reviewer is correct in his opinion in that the novelty in this paper is the finding that a simple dietary manipulation is sufficient to reduce insulin enough to inhibit tumor growth. Insulin's effect on tumor growth is known, as the reviewer points out, but the strength of our findings is that simply changing when you eat, rather than what you eat, might be enough to achieve a clinical reduction in insulin and thus slow tumor growth. The changes to the text are highlighted in yellow. We have also reformatted the figures in accordance with the journal's style. Individual data points are now shown instead of bar graphs, data availability statement added, statistics added to legends, exact p-values given for t-test (not possible for ANOVA in PRISM, post test just gives $p < 0.05$ etc), number of replicates/mice/tumors added to each legend, raw gel blot images supplied, MW indicated, and missing scale bars added. We have addressed specific comments below.

Reviewer #3 (Remarks to the Author):

The paper is improved but I still have reservations regarding its suitability for Nature Communications. The technical details are fine, but let's look at the 'big picture' and novelty. Here is a paragraph from the discussion: "In summary we found that TRF normalizes hyperinsulinemia, restores circadian rhythms, and inhibits tumor growth in models of post-menopausal obesity-driven breast cancer. These findings provide a molecular rationale for the adoption of TRF as a life-style change in humans to reduce insulin resistance and normalize circadian rhythms. Thus, TRF will likely be an effective non-pharmacological intervention to prevent or inhibit the progression of human breast cancer, and may also provide adjunctive benefit when used in combination with standard chemotherapy"

Comment 1: First, findings in murine models often do not lead to similar findings in humans, "will likely be effective" is too strong a statement - 'clinical trials to determine if the model has relevance to human cancer are justified' might be better.

Response: We have moderated the language as requested. While we agree with the reviewer that findings in murine models often do not lead to similar findings in humans, we do not believe this is the case with respect to the relationship between hyperinsulinemia and cancer based on the multitude of studies in murine models and humans demonstrating that hyperinsulinemia plays a major role in cancer progression. Furthermore, there is promising evidence from small human studies as cited in our manuscript, that TRF has beneficial metabolic effects in humans, and extensive epidemiological studies that prolonged nightly fasting, a form of time-restricted feeding, is associated with reduced breast cancer risk biomarkers and breast cancer recurrence. Thus, it is not unreasonable to suggest that TRF is likely to be an effective intervention in human breast cancer. The aim of our study was to identify the underlying mechanisms and we demonstrated that it is the ability of TRF to reduce hyperinsulinemia which drives

obesity-mediated breast cancer. While this may seem obvious in hindsight, other explanations had been proposed such as reduction of inflammation being a major driver of obesity and cancer. We agree that appropriately powered, randomized controlled clinical trials to study a role for TRF/prolonged nightly fasting in human breast cancer are a critical next step and have included this point in the discussion.

Comment 2: The authors appropriately cite prior literature (their refs 42,43,36,38) that TRF lowers insulin in mice and humans, so recapitulating this is nice, but certainly not novel, and "we found that that TRF normalizes hyperinsulinemia" might be replaced by "we confirmed and extended prior reports that TRF normalizes hyperinsulinemia".

Response: As suggested we changed the wording in our revised MS (page 21, line 569-570).

Comment 3: Finally, the notion that adverse effects of obesity on post-menopausal breast cancer are mediated at least in part by insulin is "old news" and is in the literature in many (>3000 on pubmed) papers, some cited by the authors, but many important ones and reviews not cited, including relevant Mendelian randomization studies.

Response: As the reviewer points out in comment 4, it is not the concept of insulin reduction that is interesting but the fact that TRF uses this mechanism to reduce tumor growth. Notwithstanding, we have cited many additional important articles and reviews describing the insulin driven effect on postmenopausal breast cancer in our revised MS (page 3, line75-80).

Comment 4: Certainly, if it is accepted, the discussion should be altered to be more modest and show how building on prior knowledge related to insulin and cancer (perhaps cite some of the classic and/or newer comprehensive reviews to this large field - eg: Rachel J Perry 1, Gerald I Shulman. Mechanistic Links between Obesity, Insulin, and Cancer. Trends Cancer 2020 Feb;6(2)75-78; Pollak M. The insulin and insulin-like growth factor receptor family in neoplasia: an update. Nat Rev Cancer. 2012 Feb 16;12(3):159-69) and also building on prior knowledge appropriately cited by authors regarding insulin-lowering effects of TRF, it was shown that TRF has an inhibitory effect on a murine model of breast cancer. Other ways of lowering insulin have previously been shown to have tumor inhibitory effects on various kinds of cancer in mouse models (dozens of papers, a recent example: Uncoupling Hepatic Oxidative Phosphorylation Reduces Tumor Growth in Two Murine Models of Colon Cancer. Wang Y et al Cell Rep. 2018 Jul 3;24(1):47-55.) - thus a strength of the paper is not the concept of insulin reduction, but rather the possibility that a simple diet manipulation might be the most practical way to achieve this clinically. Nevertheless, it remains completely unclear if the magnitude of the insulin reduction achieved this way would be sufficient to have a clinical impact.

Response: As suggested we have revised the manuscript to include more discussion on insulin in cancer. We have cited the important reviews mentioned above in our revised MS (page 18, line 490). Further we also edited the discussion based on prior knowledge

related to insulin and cancer and insulin-lowering effects of TRF to make it more compelling to the readers (page19, line 508-522).